METHODS AND RESOURCES

# Rewarding animals based on their subjective percepts is enabled by online Bayesian estimation of perceptual biases

Yelin Dong[1]☉, Gabor Lengyel ⓘ[1]☉*, Sabyasachi Shivkumar[1,2], Akiyuki Anzai[1], Grace F. DiRisio[1], Ralf M. Haefner[1‡], Gregory C. DeAngelis[1‡]

**1** Department of Brain and Cognitive Sciences & Center for Visual Science, University of Rochester, Rochester, New York, New York, United States of America, **2** Zuckerman Institute, Columbia University, New York, New York, United States of America

☉ These authors contributed equally to this work.
‡ These authors also contributed equally to this work.
* lengyel.gaabor@gmail.com

**Data availability statement:** The source data of the motion discrimination experiment is publicly available at https://doi.org/10.5281/zenodo.15341390 which is the first release of the reward_perception GitHub repository, https://github.com/GaborLengyel/reward_perception. The code that implements our method in the

## Abstract

Elucidating the neural basis of perceptual biases, such as those produced by visual illusions, can provide powerful insights into the neural mechanisms of perceptual inference. However, studying the subjective percepts of animals poses a fundamental challenge: unlike human participants, animals cannot be verbally instructed to report what they see, hear, or feel. Instead, they must be trained to perform a task for reward, and researchers must infer from their responses what the animal perceived. However, animals' responses are shaped by reward feedback, thus raising the major concern that the reward regimen may alter the animal's decision strategy or even their intrinsic perceptual biases. Using simulations of a reinforcement learning agent, we demonstrate that conventional reward strategies fail to allow accurate estimation of perceptual biases. We developed a method that estimates perceptual bias during task performance and then computes the reward for each trial based on the evolving estimate of the animal's perceptual bias. Our approach makes use of multiple stimulus contexts to dissociate perceptual biases from decision-related biases. Starting with an informative prior, our Bayesian method updates a posterior over the perceptual bias after each trial. The prior can be specified based on data from past sessions, thus reducing the variability of the online estimate and allowing it to converge to a stable value over a small number of trials. After validating our method on synthetic data, we apply it to estimate perceptual biases of monkeys in a motion direction discrimination task in which varying background optic flow induces robust perceptual biases. This method overcomes an important challenge to understanding the neural basis of subjective percepts.

current study is publicly available at
https://doi.org/10.5281/zenodo.15341390
which is the first release of the
reward_perception GitHub repository, https:
//github.com/GaborLengyel/reward_perception.

**Funding:** This work was supported by the
BRAIN Initiative grant from the National Institute
of Neurological Disorders and Stroke
(https://www.ninds.nih.gov/ U19NS118246 to
GCD and to RMH), and by the Computing
Module of an National Eye Institute Core grant
(https://www.nei.nih.gov/ EY001319 to GCD).
The funders had no role in study design, data
collection and analysis, decision to publish, or
preparation of the manuscript.

**Competing interests:** The authors have
declared that no competing interests exist.

## Introduction

In the natural environment, our subjective percepts often deviate from the sensory information entering our nervous system. Contextual information shapes perception of even low-level features, such as luminance [1–3], and continues to modulate perception along the processing hierarchy [4,5]. Some of the most compelling examples of biased perception are visual illusions. For example, in the classic Checkerboard illusion [6], we perceive the "white" square shadowed by the cylinder (square B in Fig 1A) to be brighter than the black square outside of the shadow (square A in Fig 1A), although the two squares have identical luminance. A systematic bias in behavior that originates from perceptual processes is called a *perceptual bias* [7]. These perceptual biases are typically quantified using decision-making tasks that measure the preference of an observer to choose one option over another (e.g., square B appears brighter/darker than square A).

Studying perceptual biases has been of great interest because it can provide insight into the underlying perceptual and cognitive processes. For example (Fig 1B), by measuring perceptual biases in motion perception, prior work has revealed that both human and animal observers judge object motion by subtracting (a portion of) the optic flow due to self-motion from the retinal flow field [8–13].

To study the neural basis of perceptual biases, animal models are particularly valuable as they provide rich electrophysiological data through invasive techniques [14,15]. However, measuring perceptual biases in animals poses a fundamental challenge [16,17]. Whereas humans can be verbally instructed to report what they see, hear, or feel without receiving feedback, animals must be trained to perform a task in exchange for some form of reward. A central challenge is that animals' behavioral reports are shaped by the reward feedback they receive [18–21]. As a result, if stimulus context biases perception away from the rewarded

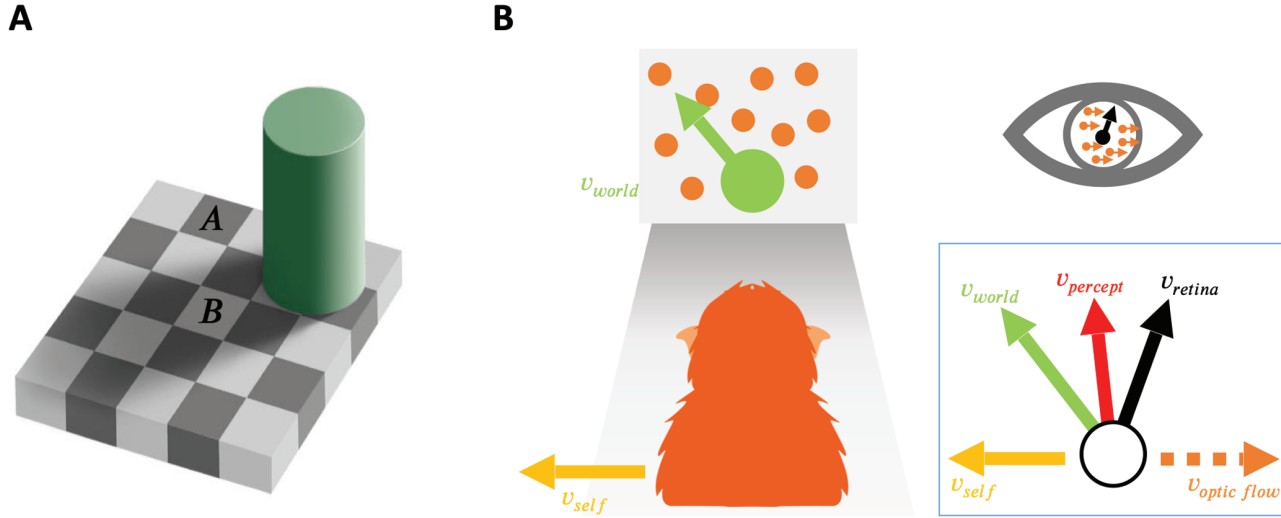

**Fig 1. Two examples of how contextual information can bias visual perception.** (**A**) Luminance illusion created by shadows [6] (source: https://persci.mit.edu/gallery/checkershadow). Square **B** looks brighter than square **A** but has the same luminance, i.e., they have identical grayscale values in the picture. (**B**) Perception of object motion is biased by self-motion [8–13]. The combination of leftward self-motion, $v_{self}$, and up-left object motion in the world, $v_{world}$, produces retinal motion that is up-right ($v_{retina}$). If the animal partially subtracts the optic flow vector (orange dashed arrow, $v_{optic\ flow}$) generated by self-motion (yellow arrow, $v_{self}$) from the image motion on the retina (black arrow, $v_{retina}$), they may have a biased perception of object motion (red arrow, $v_{percept}$) that lies between retinal and world coordinates (green arrow, $v_{world}$).

response defined in the task, animals may learn to compensate for their biased subjective percepts to receive more rewards [8,14]. This concern is especially acute for neuroscientific studies since animals often need to perform a task for many thousands of trials and rewards must be provided frequently to keep the animals motivated. Our goal here is to estimate perceptual biases online and reward animals such that they won't learn to compensate for their intrinsic perceptual biases based on reward feedback.

In previous studies, researchers followed a few different strategies for rewarding animals when the subjective percept was expected to deviate from the true stimulus value [16,17,22–34], which we shall refer to as a "bias context". In some studies, animals were always rewarded on bias context trials to ostensibly prevent the animals from compensating for their bias to receive more rewards (e.g., [27,28]). In other studies, researchers never rewarded animals on bias context trials (e.g., [24–26,35]). Other strategies involve rewarding the animal randomly (e.g., [29,30]) or with some fixed rate (e.g., [8,14,31–34]) on bias context trials, and some studies just reward animals for veridical performance, assuming no perceptual bias (e.g., [22,23]). These variations in reward strategy may lead to large individual variability in the sign, pattern, and extent of measured perceptual biases [16,17]. Some studies have highlighted in their method sections the challenge of developing effective reward strategies in bias-context trials to prevent animals from adopting compensatory response strategies (e.g., [32–34]). They acknowledged that their chosen solutions (e.g., always rewarding, never rewarding, or random rewards) are suboptimal, as these approaches can influence animals' responses and limit the number of trials suitable for studying illusions or expected perceptual biases.

Moreover, most of these previous animal studies of illusions focus on behavior, and the animals are typically only required to complete tens or hundreds of trials (e.g., [24,25,27,28, 31–34]). In such cases, animals may not learn to compensate for their perceptual biases due to limited exposure to illusion/bias trials. In many neuroscience experiments that involve electrophysiology, however, animals need to perform tasks over long periods of time (several months), often involving tens of thousands of trials. In this scenario, all of the above-mentioned approaches to rewarding animals in bias contexts become problematic, as animals have ample opportunity to learn to compensate for their perceptual biases to maximize reward. Indeed, a recent study reported that the perceptual biases of two macaque monkeys decreased over weeks and months in a motion discrimination task that invoked flow parsing [8]. Similarly, another study observed a reduction in the perceived spatial lag during flash-lag illusion trials over approximately 50 sessions in two macaque monkeys [35]. Thus, there is a critical need for an approach to estimate perceptual biases online and to reward animals around their intrinsic biases, thereby removing the drive to compensate in order to maximize reward.

We developed a method that first infers the animal's biased percept in each trial and then provides a reward based on what the animal most likely perceived. This approach requires the researcher to infer the perceptual biases of the animal online after each trial, which is challenging for the following reasons. First, in perceptual decision-making tasks, it is difficult to dissociate a perceptual bias from other decision- and response-related biases because the overall bias in the animal's responses reflects the net result of all perceptual, cognitive, and response-related processes. To disentangle these processes, previous studies attempted to dissociate the origins of response biases [36–41]. In the case of the widely-used two alternative forced-choice (2AFC) task, it is impossible to determine whether a response bias is due to perceptual bias or a decision bias [37] without using multiple different tasks [41–43], or multiple task conditions [36,38–40] that are interleaved trial by trial.

Second, even with an approach to separate perceptual biases from other decision-related biases, accurately estimating the perceptual bias from a small number of trials is difficult.

The statistically optimal method for estimating the value (and uncertainty) of a latent variable (perceptual bias) from noisy measurements (responses of the animal) is Bayesian inference. This method combines prior beliefs about the values and the likelihood of the data given the values of the latent variable [44,45]. Thus, with sufficiently informative prior beliefs, it should be possible to obtain useful estimates of the perceptual bias from a small number of trial outcomes.

In this study, we developed a Bayesian method that makes use of multiple stimulus conditions to perform online estimation of perceptual biases separately from other decision-related biases. We demonstrate the validity of our approach using ground-truth simulations and also apply it to the behavior of macaque monkeys performing a motion discrimination task. Our method allows us to estimate monkeys' perceptual biases after each trial and allocate rewards accordingly. In contrast to previous studies that used a random [8] or no [35] reward strategy in bias context trials, we show that an animal's perceptual biases remained stable across more than 50 training sessions, thus demonstrating the efficacy of our approach.

## Results

### Rewarding animal behavior relative to perceptual biases

First, we illustrate the problem of rewarding animals based on what they perceive, using the example of object motion perception in the context of self-motion (Fig 1B). Consider a task in which animals are trained to discriminate the motion direction of a patch of dots (the target) relative to a vertical direction reference. This task is tested under three contextual conditions: no self-motion, leftward self-motion, and rightward self-motion. Self-motion is simulated using optic flow, represented by a large random-dot motion background. Note that in this example, where self-motion is simulated via optic flow, object motion in the world, $v_{\text{world}}$, can be computed by subtracting the optic flow vector at the target's location, $v_{\text{optic flow}}$, from the retinal velocity of the target, $v_{\text{retina}}$.

When there is no self-motion, i.e., the animal is stationary in the world (see Fig 2A), there is no contextual information that biases perception. Thus, the perceived object motion ($v_{\text{percept}}$) matches (on average) the actual object motion in the world ($v_{\text{world}}$) and what is displayed on the screen ($v_{\text{retina}}$). In this case, if the animal is trained to perform the direction discrimination task described above, the vertical task reference (green dashed line in Fig 2A) and the motion direction that the animal perceives as vertical (red dashed line in Fig 2A) are aligned. Under these conditions, rewarding the animal is straightforward: you simply reward them in accordance with retinal or world motion (which are the same in this case). This is a common scenario in many 2AFC tasks that are performed by animals (e.g., [46,47]), in which no perceptual bias is expected such that the researcher-imposed reward boundary (blue dashed line in Fig 2A) and the task reference (green dashed line) are the same. In this case, the proportions of leftward and rightward responses are expected to be equal at the task reference direction, resulting in a psychometric curve that is centered at the vertical reference (Fig 2A, bottom).

However, it is well established that background optic flow consistent with self-motion can bias the perception of object motion [8–13]. Consider an example (Fig 2B, top) in which the animal is exposed to rightward background optic flow that simulates leftward self-translation, while the target object is moving up and to the left in the world (green vector, $v_{\text{world}}$). This combination produces image motion of the object that is up and to the right (black vector, $v_{\text{retina}}$). Previous studies have shown that the object motion perceived by humans (red vector, $v_{\text{percept}}$) typically lies between retinal velocity, $v_{\text{retina}}$, and object velocity in the world, $v_{\text{world}}$ [8–13]. If the animal was trained to discriminate motion relative to a vertical reference

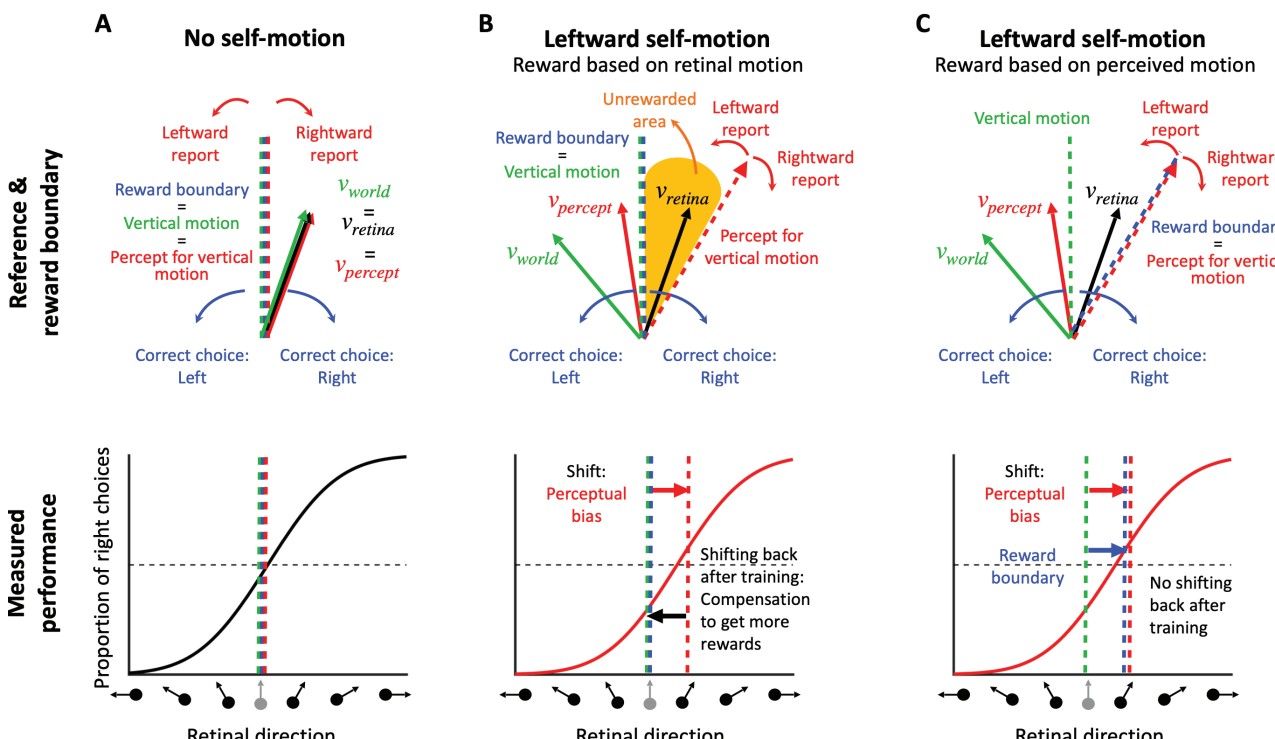

**Fig 2. Reward strategies for a motion discrimination task with simulated self-motion.** (**A**) Top: No self-motion: the perceived direction ($v_{percept}$, red arrow) matches both the retinal ($v_{retina}$, black arrow) and the world directions ($v_{world}$, green arrow). The vertical task reference (vertical motion, green dashed line) and perceived vertical motion (red dashed line) are aligned. Therefore, rewarding the animal veridically (reward boundary, blue dashed line) will not induce a perceptual bias. Bottom: The corresponding psychometric curve shows the proportion of "right" choices (y-axis) as a function of the retinal motion direction (x-axis), which equals the object motion in the world for this case. The psychometric curve shows no horizontal shift (perceptual bias, red dashed line) because the retinal, world, and perceived motion directions are the same. (**B**) Top: Leftward self-motion associated with rightward optic flow: the perceived direction ($v_{percept}$, red arrow) is likely to be shifted leftward relative to motion on the retina ($v_{retina}$, black arrow), and rightward relative to motion in the world ($v_{world}$, green arrow). If the animal is rewarded for discriminating direction relative to a vertical reference (blue dashed line), there will be a range of directions that the animal perceives as leftward but will not be rewarded (yellow area). Bottom: The psychometric curve in the leftward self-motion condition is expected to be shifted to the right, reflecting the perceptual bias of the animal (red arrow). However, with extensive training on the task, the animal is likely to adopt a compensatory strategy that shifts the psychometric curve back to the left, such that the overall response bias no longer reflects the underlying perceptual bias. (**C**) Top: If the reward boundary (blue dashed line) is rotated to match the animal's percept for vertical motion (red dashed line), then the unrewarded area is eliminated. Bottom: Using this reward strategy, the intrinsic perceptual bias of the animal can be measured from the psychometric curve even after extensive training on the task.

in screen coordinates, then flow parsing would introduce a leftward perceptual bias in behavioral reports (in retinal coordinates). This perceptual bias can be observed as a horizontal shift in the psychometric curve (red arrow in Fig 2B, bottom).

If the animal is rewarded for reporting object motion relative to the vertical reference direction (blue and green dashed vertical lines in Fig 2B, top), there will be a subset of conditions in which the animal perceives the object motion as leftward but is rewarded for choosing rightward (orange area in Fig 2B, top). To maximize rewards, the animal may learn to compensate for their perceptual bias and report a motion direction opposite to what they perceive. After an extensive training period, the shift in the psychometric curve may gradually diminish as a result of this compensation process (black arrow in Fig 2B, bottom). In extreme cases, the curve may even return to the center, thus eliminating the response bias of the animal. In this case, the researcher may incorrectly conclude that there was no perceptual bias induced by optic flow.

A solution to this problem entails rewarding animals for reporting their subjective perception rather than the veridical stimulus value in retinal coordinates. In our task example, if we shift the reward boundary to align with the animal's subjective percept of vertical motion (red dashed line in Fig 2C, top), the unrewarded area will disappear and the animal will be rewarded for reporting direction relative to its subjective vertical. With this reward boundary, rewards will not influence the animal's intrinsic perceptual bias, because the reward rate is maximized. Even after extensive training, the animal's perceptual bias is expected to persist, and thus researchers should be able to measure it as a horizontal shift in the psychometric curve (in retinal coordinates). Thus, if an animal's perceptual bias can be reliably estimated during the task, then rewarding the animal relative to their intrinsic perceptual bias should maintain stable performance over time.

As illustrated in Fig 2B, a major challenge in estimating perceptual biases is that animals may adjust their decision criteria to compensate for perceptual biases, especially if this will garner greater reward. To demonstrate the interaction between reward strategy and changing decision criteria, we simulated the behavior of a reinforcement learning (RL) agent that was given the ability to adjust its decision criteria independently for each contextual condition, based on feedback (using a temporal difference rule [48]). We compared the RL agent's behavior under various reward strategies from previous studies, including veridical rewards and three strategies for rewarding ambiguous (i.e., illusion) trials: random, always, or never. We further compare these methods to the proposed optimal strategy: rewarding based on the agent's subjective percepts. Since reward strategies differ substantially in how stimulus values are mapped to rewards (see S4 Fig), the RL agent is expected to adjust its decision criteria to compensate for any inherent perceptual biases and maximize rewards. The simulations involved a 2AFC task with three interleaved contextual conditions: one eliciting a positive perceptual bias (leftward self-motion, red lines in Fig 3 and in S3 Fig), one eliciting a negative bias (rightward self-motion, green lines in Fig 3 and in S3 Fig), and a neutral condition with no perceptual bias (no self-motion, blue lines in Fig 3 and in S3 Fig). Decision criteria were assumed to be unbiased across all conditions at the start of the simulated experiment, reflecting the common practice of training animals on the neutral condition until they achieve stable, unbiased performance before introducing contextual modulations. The RL agent adjusted its decision criteria after each trial based on reward feedback, mimicking the tendency of animals to maximize rewards (see Method section "Reinforcement learning with different reward methods" for details).

The simulations clearly demonstrate that conventional reward strategies (veridical and random rewards) lead to an underestimation of perceptual biases that grows more severe over time (Fig 3B, 3D and S3 FigB, S3 FigD). This underestimation occurs because the RL agent adjusts its decision criteria in response to feedback, compensating for its inherent perceptual biases to maximize rewards (Fig 3A, 3C and S3 FigA, S3 FigC). For example, when rewards are based on the veridical stimulus value—effectively ignoring the possibility of biased perception—the measured biases disappear completely after about 20 sessions (Fig 3B, note that the exact time course will depend on the learning rate of the RL agent). Similarly, when rewards are allocated randomly (Fig 3C, 3D), always (S3 FigA, S3 FigB), or never (S3 FigC, S3 FigD) in stimulus conditions for which the expected percept is ambiguous, measured biases decrease substantially over 20-30 sessions before stabilizing at a reduced plateau level, similar to findings in previous empirical studies with animals (e.g., [8,35]). These results demonstrate clearly that perceptual biases cannot be measured accurately using previous reward strategies if animals are capable of adjusting their decision criteria in a context-specific manner.

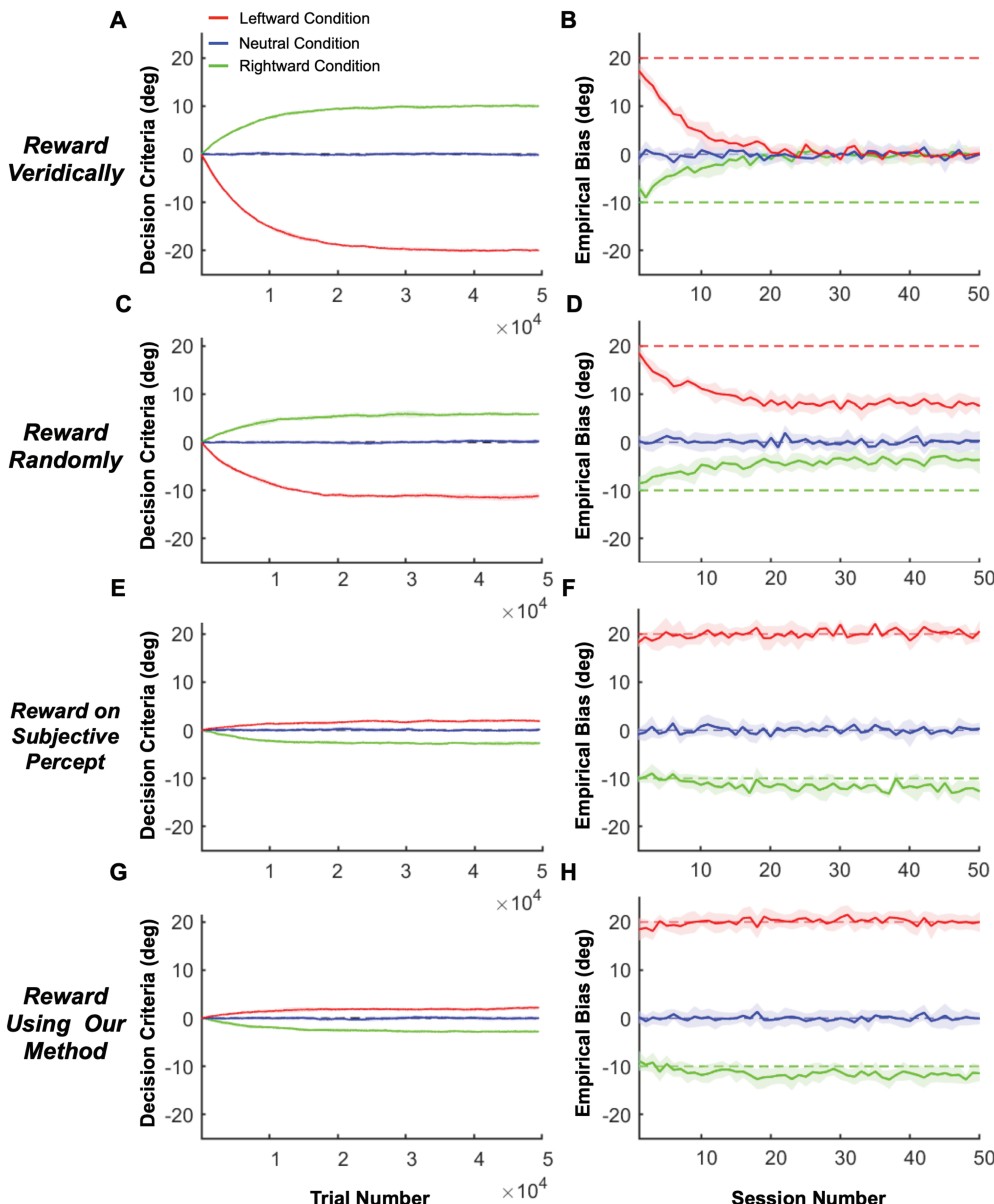

**Fig 3. Interactions between reward strategy and decision criteria in simulations of an RL agent.** (**A, B**) Rewarding the RL agent based on veridical stimulus values. (A) The mean (solid curves) and SD (shaded areas) of the learned decision criteria across 10 simulations involving 50 sessions of 990 trials each (990 x 50 = 49500 trials, x-axis). Red and green: leftward and rightward self-motion conditions, respectively. Blue: neutral condition with no self-motion. The dashed black line (barely visible behind the solid blue curve) represents a decision criterion of zero. (B) The mean (solid lines) and SD (shaded areas) of the estimated empirical biases in the same three conditions (format as in A) for each simulated session (990 trials per session). Ground truth empirical bias was +20 in the leftward self-motion condition (dashed red) and -10 in the rightward self-motion condition (dashed green). (**C, D**) Same as (A, B), but allocating rewards randomly in ambiguous trials. (**E, F**) Same as (A, B), but allocating rewards based on the ground truth perceptual biases (i.e., subjective percept). (**G, H**) Same as (E, F), but using our method to estimate the subjective perceptual biases from the agent's responses and allocate rewards based on those estimates. Source data for Fig 3A–3H are available in "Figure3&FigureS3.mat" at https://doi.org/10.5281/zenodo.15341390.

In contrast, when rewards are allocated based on the true perceptual biases for each context (i.e., the reward boundary matches the inherent perceptual biases), the RL agent barely adjusts its decision criteria (Fig 3E) and the estimated empirical biases remain stable over time, matching the true perceptual biases (Fig 3F). Note that decision criteria for the leftward and rightward conditions deviate slightly from zero even when using this optimal reward strategy due to the uneven distribution of stimulus values corresponding to each choice (see Method section "Reinforcement learning with different reward methods" and S3 FigE, S3 FigF for details). However, this deviation is minimal compared to other reward strategies and disappears if stimulus values are symmetrical around the perceptual bias for each contextual condition (S3 FigE, S3 FigF).

In real experiments, researchers lack access to the true biases of animals. To address this, we propose a method that estimates perceptual biases online based on the animal's response to each trial, enabling reward allocation to be guided by these estimates. As shown in Fig 3G, 3H, our method—detailed in the next section—produces results that are nearly indistinguishable from the optimal strategy, demonstrating that it can accurately estimate and stabilize perceptual biases in tasks with multiple interleaved contextual conditions.

## Disentangling perceptual and decision biases using a Bayesian approach

In general, perceptual biases cannot simply be measured as the shift of the psychometric curve. Since a psychometric curve reflects both perceptual and decision-related processes, attributing the cause of the shift only to biased perception is impossible in most typical 2AFC tasks [37,49–52]. We refer to the measured horizontal shift of the psychometric curve as the "empirical bias" (denoted by $B$ in Fig 4A), and we divide it into two components: (1) perceptual bias and (2) decision bias (see $P_L$, $P_R$, and $D$ in Fig 4B). We refer to all biases that are not related to how the animal *perceives* the stimulus as "decision biases," which includes any biases in decision-making and motor planning/execution. For example, the decision criteria the RL agent adjusted in our simulations in the previous section (Fig 3) contribute to the decision bias. Since the empirical bias reflects a combination of perceptual and decision biases, it is impossible to separate them using a simple 2AFC task (Fig 4A).

However, if we have at least two experimental conditions, we can disentangle perceptual and decision biases (two unknowns) if we know how the biases in the two conditions relate to each other (two constraints) (e.g., see [36]). For instance, when multiple stimulus conditions are interleaved and the animal has no reward-based incentive to adopt different decision criteria across conditions, we can reasonably assume that the decision bias is the same across these conditions. In this scenario, the RL agent would also not develop separate decision criteria, as rewards can be maximized using the same criterion across conditions (Fig 3E, 3G). This aligns with standard practices, in which animals are initially trained extensively under a neutral condition to achieve minimal bias and stable performance before introducing contextual modulations. When contextual modulations are introduced, if rewards are allocated to align with inherent perceptual biases, there is no incentive for the animal to develop different decision criteria across conditions (Fig 3E, 3G). This strategy will be effective provided that animals learn to adjust their decision criteria more slowly than our method can estimate their perceptual biases to allocate rewards (see related simulations in S5 FigC, S5 FigD). However, if the animal can adjust its decision criteria more rapidly than our algorithm converges to the true perceptual biases, it may be impossible to prevent the development of separate criteria across conditions (see related simulations in S5 FigA, S5 FigB). While other types of decision biases not related to reward maximization could emerge, there is no reason to assume

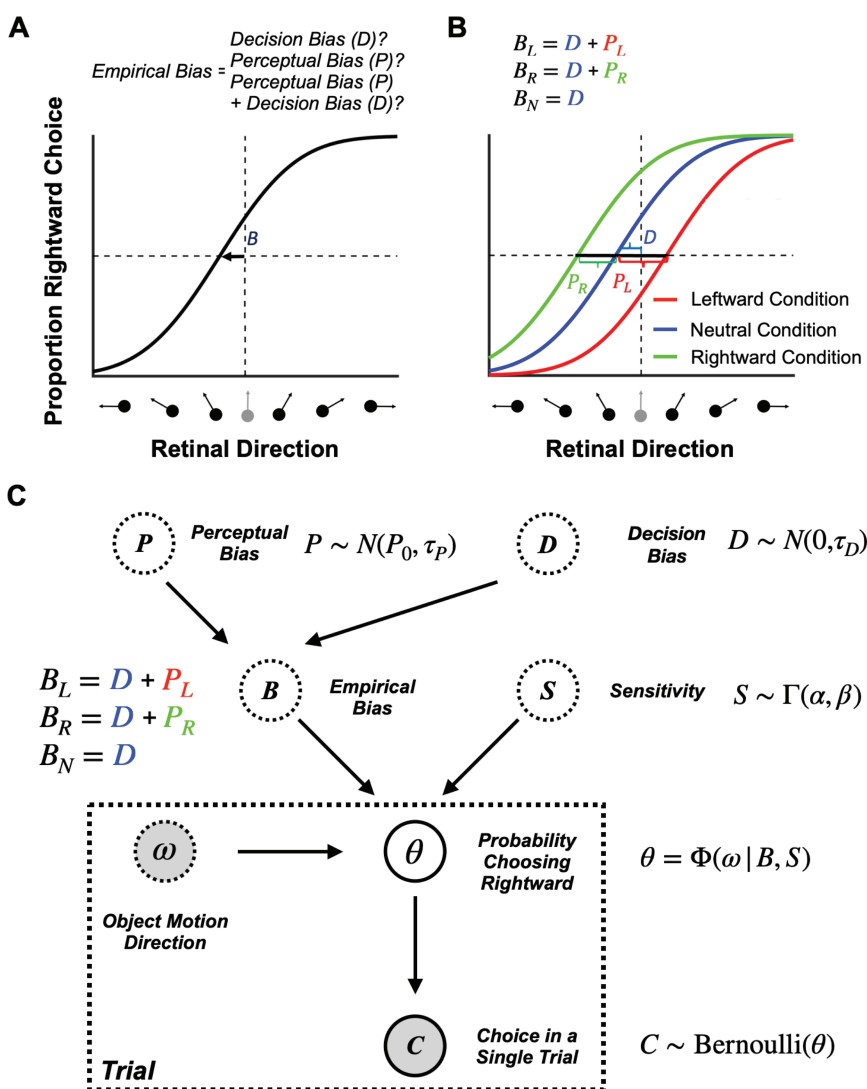

**Fig 4. Disentangling perceptual and decision biases.** (**A**) Both perceptual and decision-related biases can shift the psychometric function horizontally in 2AFC tasks. From a single psychometric curve, it is impossible to know whether the empirically measured shift, (*B*), was a decision bias, a perceptual bias, or a combination of the two. (B) Separating perceptual and decision biases with multiple stimulus conditions in the case of judging object motion during self-motion. Equations show how to compute the perceptual ($P_R$ and $P_L$) and decision (*D*) biases from the empirically measured biases ($B_L, B_R$ and $B_N$) in three stimulus conditions with leftward, rightward, and no self-motion, respectively. We assume that the decision-related bias (*D*) is constant across the three interleaved conditions. (**C**) The generative model of our Bayesian approach for estimating perceptual and decision biases. The animal's choice in each trial, *C*, follows a Bernoulli distribution with $\theta$ denoting the probability of choosing rightward motion relative to the reference. The psychometric curve, $\Phi$, reflects the relationship between $\theta$ and object direction, $\omega$, as captured by a cumulative Gaussian distribution. The sensory noise *S* and the empirical bias *B* influence the slope and shift of the psychometric curve, respectively. Empirical bias for each condition *B* is determined by the perceptual bias *P* and decision bias *D* variables. $\Gamma(\alpha, \beta)$, denotes a gamma distribution while $N(P_0, \tau_P)$ and $N(0, \tau_D)$ denote normal distributions. Note that we also account for lapse rates, though they are omitted from this figure for simplicity. See Method section "Hierarchical Bayesian model" for further details.

that such biases would differ across the interleaved contexts. Next, we describe three experimental designs in which it is possible to estimate perceptual biases separately from a common decision bias.

First, consider an experiment with two conditions in which the context-induced perceptual biases are expected to have equal magnitudes, but opposite signs. Then, subtracting the empirical biases measured in these two conditions will give an estimate of the perceptual bias that is not dependent on the common decision bias across the two conditions.

Second, consider an experiment with a single contextual modulation that induces a perceptual bias, as well as an interleaved neutral control condition for which no perceptual bias is expected. In this case, the empirical bias measured in the control condition estimates the animal's decision bias, which can then be subtracted from the empirical bias in the contextually modulated condition to obtain the perceptual bias.

Third, we consider an experiment with three conditions: one in which context induces a leftward perceptual bias, another in which it induces a rightward perceptual bias (not necessarily of the same magnitude), and a third neutral condition for which no perceptual bias is expected. This is the scenario in which we present our method here. In the neutral condition, the empirical bias reflects only the decision bias. The two perceptual biases can then be computed by subtracting the empirical bias measured in the neutral condition from the empirical biases measured in the other two conditions (Fig 4B). Importantly, our method leverages this idea conceptually but, in practice—details provided in the next section—it jointly infers a common decision and two separate perceptual biases across all three contextual conditions. This approach is broadly applicable to any 2AFC task with different contextual modulations that induce perceptual biases. An advantage of using three conditions is that researchers do not have to assume that perceptual biases have equal magnitudes and opposite signs; their magnitudes can vary freely.

To implement this method with three contextual conditions in the motion discrimination task example (Figs 1 and 2), we interleave trials with leftward self-motion, no self-motion, and rightward self-motion modulations. Based on prior studies [8–13], we expect opposite-sign perceptual biases for the two self-motion conditions, and no bias in the neutral (no self-motion) condition. Assuming no reward-based incentives to promote distinct decision criteria, the animal's decision bias is likely to remain consistent across these interleaved conditions.

Building on this conceptual approach, we have developed a Bayesian model to estimate perceptual biases online during 2AFC tasks (Fig 4C, and see Method section "Hierarchical Bayesian model"). We model the decision-making process in the 2AFC task as follows. First, as commonly done (e.g., [53,54]), we assume that the animal's choice on each trial, $C_t$, (i.e., rightward or leftward response) follows a Bernoulli distribution with $\theta$ denoting the probability of choosing the rightward response alternative. Second, similar to most previous methods (e.g., [8,36,53]), we used a cumulative Gaussian distribution as the functional form of the psychometric curve, reflecting the relationship between $\theta$ and the stimulus value, $\omega$ (e.g., object motion direction). The two important parameters of the psychometric curve are (1) the sensory noise, $S$, which controls the slope of the psychometric curve (i.e., how rapidly $\theta$ changes as a function of $\omega$), and (2) the empirical bias, $B$, which controls the horizontal shift of the psychometric curve (i.e., where the proportion of the two response alternatives are equal).

Crucially, we further assume that the empirical bias, $B$, reflects the sum of decision ($D$) and perceptual ($P$) biases. As mentioned earlier, we assume that all three conditions share the same decision bias; however, the perceptual bias is assumed to differ across self-motion conditions (Fig 4C). Using a Bayesian framework, by selecting appropriate prior distributions for all root latent variables, including the perceptual biases, $P_L$ and $P_R$, in the leftward and rightward self-motion conditions, a common decision bias, $D$, and three separate sensory noise variables $S_L$, $S_N$, and $S_R$, for the three self-motion conditions, we can achieve accurate estimates of the perceptual biases using a small number of trials. As more trials are added, the

estimates of the biases become increasingly more accurate, as demonstrated below using synthetic data (Fig 5). Note that this model is designed to estimate the average biases within a session. Consequently, if biases fluctuate during a session, the algorithm returns their average values.

Importantly, the use of an informed prior (based on past experiments) allows us to reward the animal based on an estimate of their perceptual bias from the beginning of the experiment, whereas estimates of the bias based on a small number of trials would be wildly inaccurate without such a prior. At the start of each (simulated or real) session, rewards were based solely on the prior estimates. After collecting data from the first 33 trials—one data point for all 33 unique stimuli—we used Bayesian updating to combine the behavioral choices from these trials with the method's prior beliefs. This allowed us to compute an initial estimate of the posterior distribution over perceptual and decision biases. Then, we update our estimates of the posterior distributions over each bias based on the animal's response in each subsequent trial. To perform inference in our Bayesian model, we used Gibbs sampling, which converged within 1–3.5 seconds, even on old computer hardware used for the animal behavioral experiments (see Method section "Hierarchical Bayesian model" for further details and a formal description of the model).

After estimating the posterior distributions for each bias after each trial, we align the reward boundary with the mean of the inferred posterior distributions over the perceptual bias for conditions with contextual modulations, i.e., the leftward and rightward self-motion conditions (see Method section "Reward allocation" for details on reward allocation).

## Validation of the algorithm using synthetic data

To assess the validity of our method for estimating perceptual biases online in 2AFC paradigms, we generated synthetic datasets simulating training sessions in the motion discrimination task with leftward self-motion, no self-motion, and rightward self-motion conditions. In line with the assumptions of our Bayesian model (see Method section "Model validation" for more details), all biases were assumed to be stationary within a session. We generated 100 synthetic sessions with different perceptual and decision biases, each with 990 trials.

Consider a simulated dataset in which there are asymmetric perceptual biases for leftward and rightward self-motion, as well as a substantial decision bias (Fig 5A). Our approach yielded estimates of both perceptual and decision biases that fluctuated within a reasonably narrow band around the ground truth values (Fig 5A, dashed lines). As expected, the uncertainty bands around our estimates shrank continuously as the number of trials increased (Fig 5A). Next, we quantified the accuracy of our method for three scenarios (Fig 5C): in the first ("lucky") one (light green), the ground truth perceptual bias coincides with the mean of the perceptual prior; in the second (typical) case (medium green), the ground truth is 1 standard deviation (SD) away from the prior mean, and in the third ("unlucky") case (dark green), the ground truth is 2 SDs away from the prior mean. For clarity, we present data only for the rightward self-motion condition (Fig 5C); results for the leftward self-motion condition are analogous. As expected, the root mean square error (RMSE) grew with the mismatch between prior expectations and ground truth. Importantly, for the typical scenario, the RMSE was substantially lower than the error obtained by assuming a flat prior (i.e., uniform prior, equivalent to maximum likelihood estimation, Fig 5C, black) throughout the entire session. Additionally, for all of these cases, our method produces much less variable estimates of perceptual

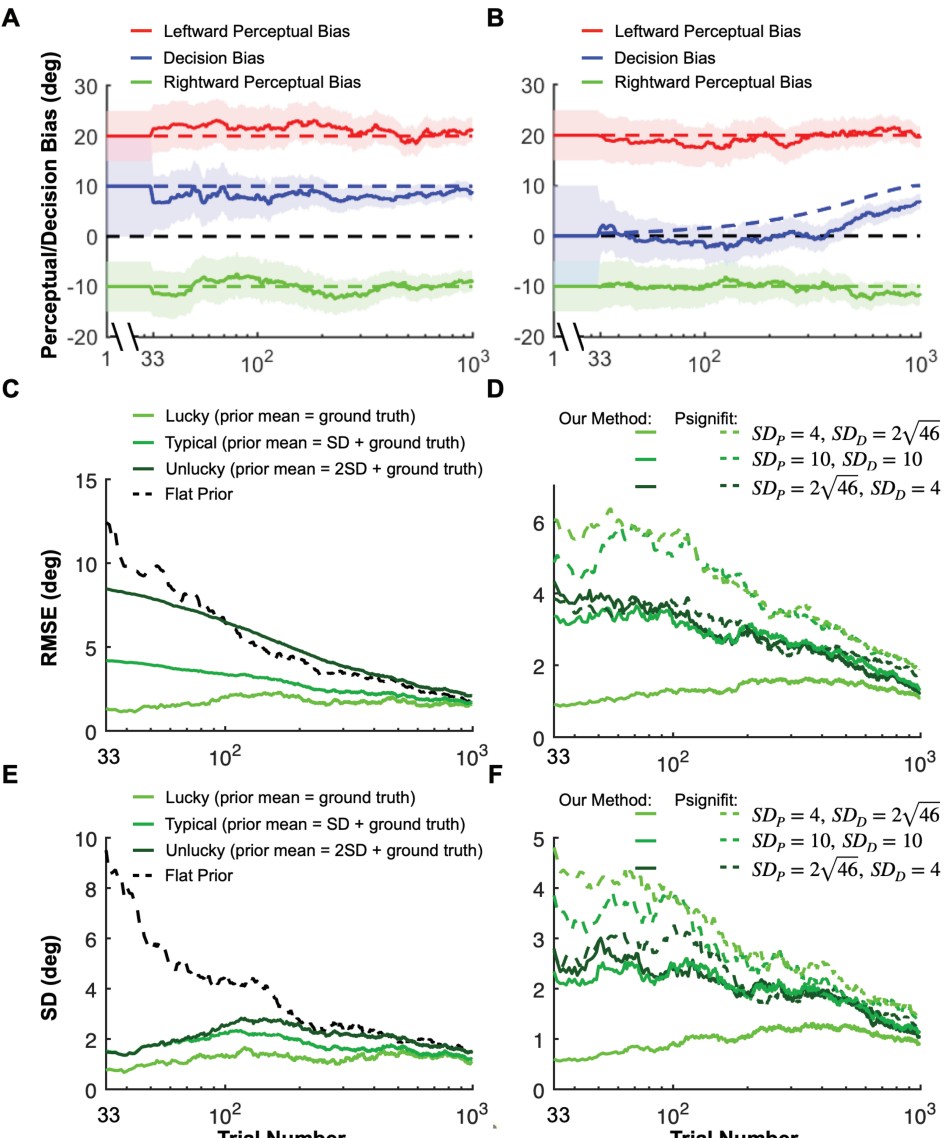

**Fig 5. Validating our method with ground truth simulations.** (**A**) The mean (solid curves) and the standard deviation (SD, shaded areas) of the inferred perceptual (red in the rightward and green in the leftward self-motion conditions) and decision (blue, neutral, no self-motion condition) biases for an example synthetic data set. Ground truth perceptual bias was +20 in the leftward self-motion condition (dashed red) and –10 in the rightward self-motion condition (dashed green). The decision bias common to all conditions was +10, as seen in the neutral, no self-motion condition (dashed blue). Before one data point was collected for each unique stimulus (first 33 trials), posterior distributions over the biases were not yet available. Therefore, we show prior distributions in this interval, with the mean (solid lines) and standard deviation (shaded areas) depicted. (**B**) The same as (A), but showing an example synthetic dataset with a slowly changing decision bias. (**C**) Average root mean square error (RMSE, y-axis), across 100 simulations, in estimating perceptual bias in the rightward self-motion condition, plotted as a function of trial number. Results are shown for three different prior mean values: 0, 1, and 2 SDs away from the ground truth perceptual bias (from light to dark green, respectively). The black curve demonstrates results for a maximum likelihood estimator which corresponds to a Bayesian estimator with a uniform prior. (**D**) Average root mean square error (RMSE, y-axis), over 100 simulations, in estimating perceptual bias in the rightward self-motion condition for three different values of prior widths for perceptual and decision biases (solid curves, light green to dark green, respectively). See text for details. Dashed curves show analogous results obtained using the conventional Bayesian Psignifit library. (**E**) The same simulation as in (C), but the SD of the perceptual bias averaged over 100 simulations is plotted. (**F**) The same simulation in (D), but the SD of the perceptual bias over 100 simulations is plotted. Source data for Fig 5A–5F are available in "simulation_data/Figure5A.mat"—"simulation_data/Figure5DF_stan.mat" at https://doi.org/10.5281/zenodo.15341390.

bias, as compared to using a flat prior (Fig 5E), especially over the first 100 trials of a simulated session. The low variability of the estimates is crucial such that the reward boundary does not fluctuate wildly across trials.

A key difference between our method and state-of-the-art methods available through off-the-shelf libraries, such as Psignifit [53], lies in our model's ability to decompose empirical biases into distinct decision and perceptual biases. This separation allows us to apply separate prior distributions to each type of bias, reflecting finer-grained knowledge from previous sessions or other subjects, for example. In contrast, employing a Bayesian method from an off-the-shelf library confines one to assigning priors solely to the empirical biases. Therefore, we assessed under what circumstances and to what extent our method outperforms the conventional Bayesian approach, as implemented using the Psignifit library [53]. To ensure comparability between the two approaches, we aligned the prior distributions over empirical biases. Decision and perceptual biases are modeled as independent, normally distributed random variables, with their sum representing the empirical bias. Using the closed-form solution for the sum of two independent Gaussian random variables, we computed the prior distributions over empirical biases based on the prior distributions of decision and perceptual biases in the leftward and rightward conditions. After estimating the posterior distributions over empirical biases using Psignifit, we recovered the perceptual biases by leveraging the normality assumption and treating the empirical bias in the neutral condition as the decision bias. This approach allowed us to directly compare the accuracy of Psignifit and our method in estimating perceptual and decision biases.

We performed simulations with three sets of prior distributions over the decision and perceptual biases (Fig 5D, 5F). The mean values of these priors were consistently aligned with the actual ground truth. However, we varied the SD of the perceptual and decision biases across three scenarios. Initially, we set a broad SD for perceptual biases ($SD_P = 2\sqrt{46}$ degrees) and a narrower one for the decision bias ($SD_D = 4$ degrees) (dark green in Fig 5D, 5F). The second scenario involved equal SDs for both types of biases (10 degrees) (medium green in Fig 5D, 5F). In the final case, the roles were reversed, with a narrow SD for perceptual biases ($SD_P = 4$ degrees) and a broader one for decision bias ($SD_D = 2\sqrt{46}$ degrees) (light green in Fig 5D, 5F). Crucially, despite these variations in the width of the prior distributions, the empirical biases for both rightward and leftward conditions remained the same, centered around the ground truth values with an SD of $10\sqrt{2}$ degrees. Therefore, the conventional Bayesian methods (which we simulated with Psignifit) would yield similar estimates of empirical biases and their uncertainties across all scenarios both for leftward and rightward conditions. There will still be direct information about the decision bias from the neutral condition even when using conventional Bayesian methods (i.e., Psignifit). Since the SD of the decision bias is different across the three scenarios it will affect the estimation of the perceptual bias. Nevertheless, we expect that our method will outperform conventional Bayesian methods (i.e., Psignifit) when there is an informative prior on the perceptual biases. Indeed, we found that both the RMS errors (Fig 5D) and the standard deviations (Fig 5F) of the perceptual bias estimates were substantially lower for our method than for conventional Bayesian methods (i.e., Psignifit), especially for the first 200 trials. The errors for our method were especially low in the case where the priors over the perceptual biases are narrow relative to the decision bias (light green curves). Consequently, our method has a considerable advantage over conventional Bayesian methods when the experimenter has a well-informed prior belief about the perceptual biases but does not know the decision-related bias of the animal before the training session.

We also tested how robust our method is when one of the assumptions of our Bayesian model is violated. Specifically, we tested scenarios in which only the perceptual biases were

stationary over time, while the decision bias changed slowly within a session (Fig 5B). Interestingly, our method was robust against the slowly changing decision bias. The perceptual biases were estimated as accurately in the changing decision bias dataset as in the stationary decision bias dataset, and only the decision bias was systematically underestimated (Fig 5B).

Furthermore, we rewarded the RL agent using our Bayesian method and we compared the results to simulations in which rewards were based on the ground truth perceptual biases. As shown in Fig 3G, 3H, results from our method are nearly indistinguishable from the optimal strategy of using the ground truth perceptual biases (Fig 3E, 3F), demonstrating that it can accurately estimate and stabilize perceptual biases in 2AFC tasks.

## Application to monkey behavioral data

We applied our method to reward a monkey during training in an experiment investigating motion perception with self-motion simulated by optic flow. The task of the monkey was to decide whether a patch of dots, referred to as the target, was moving rightward or leftward with respect to an implicit reference (Fig 6A). We generated optic flow to simulate self-motion by displaying a full-field random-dot motion background. The discrimination boundary for our discrimination task (white dashed line in Fig 6A) was aligned with the optic flow vector at the target's location during simulated straight-forward translation (Neutral condition, Fig 6A, center).

To estimate perceptual biases separately from decision-related biases using our method, the experiment involved three interleaved conditions (as suggested in Section "Disentangling perceptual and decision biases using a Bayesian approach"). In the Neutral condition, optic flow simulated straight-forward translation, such that we expected responses to be unbiased (Fig 6A, center). In the other two conditions, optic flow simulated slightly different heading directions (red and green circles, Fig 6A, left and right), such that the optic flow vector at the location of the target would be slightly leftward (red) or rightward (green) of the discrimination boundary. Unlike the stimulus conditions that elicit optic flow parsing (Fig 1B), here optic flow induces an attractive perceptual bias rather than a repulsive bias, presumably because the optic flow is more closely aligned with the target motion. For our purposes here, we simply use these task conditions to illustrate the application of our method.

We trained one monkey to perform the motion discrimination task while using our method to estimate perceptual biases and deliver rewards. The monkey showed a leftward perceptual bias in the Leftward condition and a rightward perceptual bias in the Rightward condition (i.e., attractive biases). Psychometric curves for two example sessions are shown in Fig 6B, illustrating the average empirical response biases using data from the entire session. The online estimates of perceptual and decision biases across trials in the same example sessions, obtained using our method, are shown in Fig 6C. We used priors over the perceptual biases computed from data obtained in previous sessions (discussed further below). For the decision bias, we always used a prior centered at zero with a standard deviation estimated from previous sessions. Note that the perceptual and decision biases shown in Fig 6B correspond to the final values in the time series shown in Fig 6C. This correspondence is particularly evident for the decision bias, as it directly corresponds to the empirical bias in the psychometric curve for the neutral condition (blue lines).

In the first example session (Fig 6C, top), the monkey's perceptual biases turned out to be substantially greater than the prior mean (compare starting and ending values on the y-axis). Nevertheless, our Bayesian method quickly converged to a stable estimate of the perceptual biases, which enabled a stable reward schedule for the monkey with modest trial-to-trial variability. In the second example (Fig 6C, bottom), we can observe a situation in which the

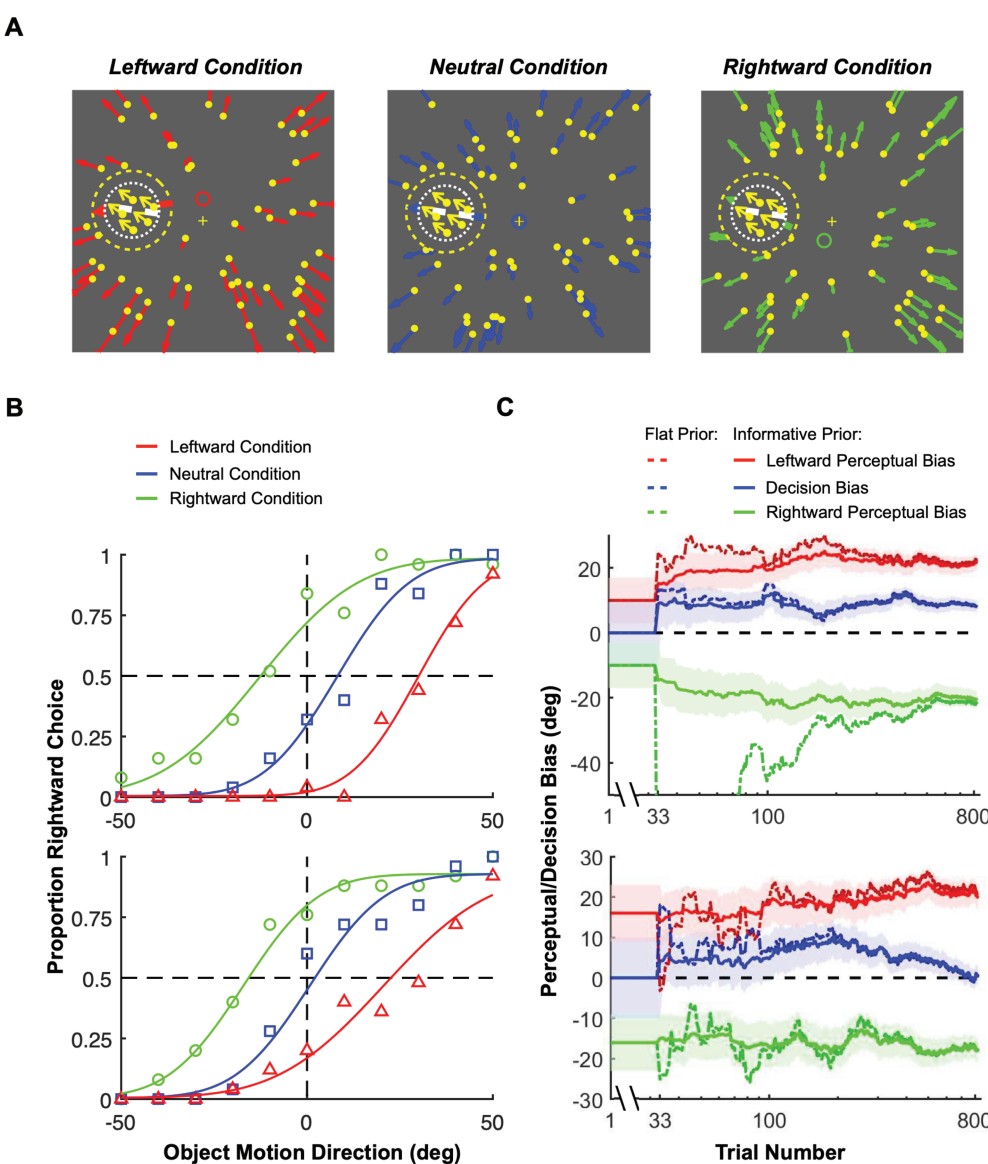

**Fig 6. Applying our method to train monkeys in a motion discrimination experiment.** (**A**) Three experimental conditions with optic flow fields that simulated different forward self-motions. Yellow dots with colored arrows represent the background optic flow vectors. The dashed inner circle shows the location of the target patch, which contains moving dots (yellow dots and arrows). The dashed outer circle represents a mask region within which no background dots appear. White dashed lines represent the implicit task reference around which the monkey had to discriminate the direction of the target patch. The small blue, red, and green rings represent the focus of expansion for each of the three optic flow fields. Neutral condition: heading direction is forward (blue ring). Leftward condition: The heading direction is slightly upward (red ring), such that the optic flow vector at the target location is leftward of the reference. Rightward condition: The heading direction is slightly downward (green ring), such that the flow vector at the target location is rightward of the task reference. (**B**) Psychometric functions from two example sessions (top and bottom), color-coded as in panel A. Smooth curves show fitted psychometric functions. (**C**) The mean (solid curves) and the uncertainty (68% CI, shaded areas) of the inferred posteriors over perceptual (red, leftward condition; green, rightward condition) and decision (blue, neutral condition) biases across trials for the same two example sessions, using informative priors. Dashed lines: estimation using flat priors. We truncated the y-axis of the upper panel because the flat prior estimation for rightward perceptual bias has large negative values. When using flat priors, estimates of the perceptual and decision biases show much larger fluctuations over the first few hundred trials (dashed curves). Reward was delivered based on the means inferred using informative priors. Source data for Fig 6B-C are available in "analysis_data/Figure6B1.mat"—"analysis_data/Figure6C2.mat" at https://doi.org/10.5281/zenodo.15341390.

decision bias of the monkey appears to change within the session. As shown in ground-truth simulations, our method appears robust against such changes in decision bias (see Fig 5B).

## Integrating hyperpriors to combine multiple sessions with varying experimental variables

In the previous sections, we showed that our method can provide an accurate online estimation of perceptual and decision biases within a session. However, the performance of the algorithm is directly related to the strength of prior beliefs about the perceptual and decision biases. Had we used uninformative, uniform priors for estimating the biases, we would have observed substantial fluctuations in bias estimates over the first few hundred trials (e.g., dashed lines show larger fluctuations than solid lines for both examples in Fig 6C). Rewarding animals based on such wildly varying estimates of perceptual bias may confuse the animals and impair the progression of training. Therefore, it is important to understand how prior beliefs influence bias estimation in our method.

To illustrate the consequences of being overconfident or underconfident, we generated another set of synthetic data with fixed perceptual and decision biases but with priors that have varying widths and fixed means centered on values that are 10 degrees away from the ground truth values. Specifically, the ground truth biases are $P_L$ = 20, $P_R$ = −20, and $D$ = 10 in this simulation (dashed lines in Fig 7A), whereas the corresponding prior means are 10, −10, and 0 (colored circles at zero in Fig 7A), respectively. The results of this simulation show the classic phenomenon of bias-variance trade-off in statistics: narrower (overconfident) mismatched priors lead to low-variability estimates that are biased away from the true perceptual bias values, while wider (underconfident) mismatched priors result in less biased but highly variable estimates (Fig 7A). Note that the error bars in Fig 7A represent variability in the estimated mean biases across multiple simulated training sessions (with each simulated session having a different stochastic sequence of simulated choices), not uncertainty around the bias as in our other figures. Therefore, using well-chosen priors that are based on posterior beliefs after observing data from previous sessions is important for achieving a good balance between variability and accuracy.

Fortunately, our Bayesian method provides an optimal framework to combine data across sessions. Moreover, if some experimental variables that influence perceptual biases change between sessions (but remain constant within a session), we can extend our model by incorporating a hyperprior for these task variables. Taking the previously described motion direction discrimination task as an example, we observed that perceptual biases depend roughly linearly on heading direction (the focus of expansion of the optic flow), and the eccentricity of the target location (S2 Fig). Therefore, we modeled the priors over the perceptual biases for a session as Gaussian distributions centered around a weighted linear combination of heading direction and eccentricity. The weights in the linear combination can then be inferred with a hierarchical Bayesian model using data from multiple previous sessions with different heading directions and eccentricity values (see Method section "Extended hierarchical Bayesian model," S1 Fig, and S2 Fig for more information). Thus, we use the information obtained from all previous training sessions, even though values of heading direction or eccentricity vary across sessions.

This approach allows us to select priors with optimal widths inferred from previous sessions for the online estimation in each subsequent session. As more sessions are completed, the uncertainty in the priors decreases and eventually converges to values that presumably reflect both the limited explanatory power of our simple linear model and any intrinsic variability in the animal's perceptual biases from day to day that is not under experimental control

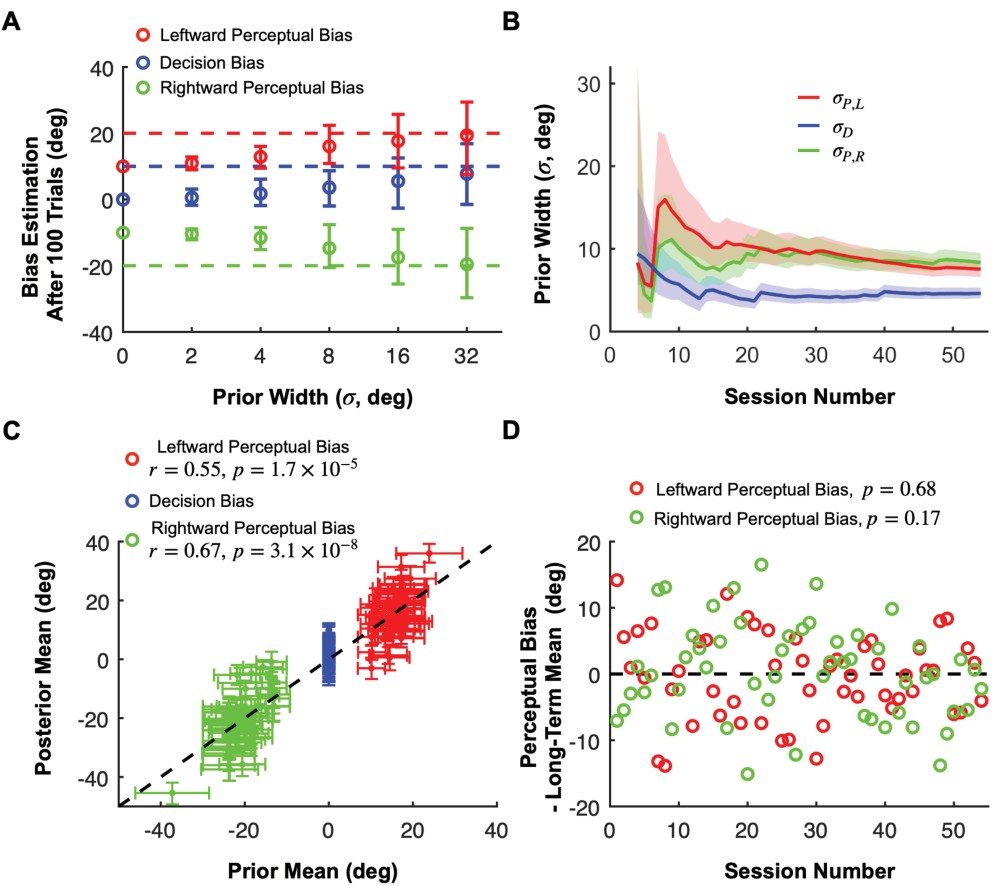

**Fig 7. Importance of priors and hyperpriors in our method.** (**A**) Demonstrating how prior width influences the mean and variance of estimated bias values. The perceptual (red and green) and the decision (blue) biases were estimated after 100 trials in synthetic sessions with fixed ground truth biases (dashed colored lines) using priors with different widths (x-axis) but fixed means (colored circles at 0 width). The circles and error bars represent the means and the SDs across 20 simulations for each prior width, respectively. (**B**) Dynamic narrowing of the estimated prior widths for the perceptual (green and red) and decision biases (blue) as the multi-session linear model integrates datasets from a progressively greater number of sessions (in chronological order). Solid lines and shaded areas represent the median and SD, respectively, of the prior widths for the perceptual and decision biases. (**C**) The relationship between the prior means and the posterior means of the perceptual (green and red) and decision biases (blue). The prior means are estimated from the linear model using data from all sessions whereas the posterior means are inferred after integrating the monkey's choices in each session. (**D**) The differences between estimated priors and posteriors of perceptual biases across training sessions. Data are shown separately for leftward (red) and rightward (green) self-motion conditions. The data suggest that perceptual biases remain consistent throughout training with our reward method. Source data for Fig 7A–7D are available in "simulation_data/Figure7A.mat" and in "analysis_data/Figure7BCD&FigureS2.mat" at https://doi.org/10.5281/zenodo.15341390.

(Fig 7B). This provides us with a calibrated measure of uncertainty that is neither under- nor over-confident.

When applying our method to real data, the ground truth biases are unknown, unlike in simulations, which prevents us from demonstrating the convergence of the estimates to the true biases of the animal. However, after applying a hierarchical multi-session model to all training sessions, we analyzed how well the inferred linear model can be used to estimate the biases in each session. We used the linear weights from the extended model, estimated using

data from all sessions, to compute the mean of the perceptual bias prior for each session, representing our estimated perceptual bias before the session begins. Across sessions, these prior estimates (shown on the x-axis in Fig 7C) are positively correlated with the posterior estimates of perceptual biases (Pearson correlation; leftward perceptual bias: $r = 0.55$, $p = 1.7 \times 10^{-5}$; rightward perceptual bias: $r = 0.67$, $p = 3.1 \times 10^{-8}$), which are updated after integrating all of the choices made by the monkey within each session (y-axis in Fig 7C). This means that our linear model for combining data across sessions accounts for about a third of the variance in perceptual biases (also see S2 Fig). The remaining unexplained variance is due to variables other than eccentricity and heading direction that are either knowable to the experimenter and could therefore be included in an improved model, or are unknowable such as internal brain states that appear as random variability. Importantly, our Bayesian model accounts for these unknowable factors by the width of the prior, representing our lack of knowledge about the true perceptual and decision biases before the session (Fig 7B). Due to this unexplained variability, the prior widths of the biases converge to fixed values (5–10 degrees) rather than decreasing steadily.

Interestingly, this property of the priors gives our method the flexibility to estimate and track perceptual biases that vary within and across sessions. For example, in the session illustrated in Fig 6C (top), the prior mean (initial y-axis value) was substantially lower than the monkey's perceptual bias for that session. Despite this initial discrepancy, our method quickly converged (within 200 trials) to the animal's presumed true perceptual bias. With smaller prior widths, the convergence would have taken longer. This effect, similar to what is shown in Fig 7A, highlights how prior width influences bias estimation since there will always be a mismatch between the true bias and the prior mean which is the basis for the reward during the first few trials in a session. To further explore the algorithm's ability to track changes in perceptual biases, we simulated the RL agent's performance in scenarios for which the ground-truth perceptual biases changed gradually over time. As described in S5 Fig, our method is very effective in tracking slow changes in perceptual biases as long as the agent does not have independent decision criteria that can change rapidly (see also Discussion).

Regarding the decision bias, we hypothesized that it is not influenced by task parameters such as heading direction or eccentricity. Thus, we always used a prior centered around zero for the decision bias with a variance reflecting its session-to-session variability. We found that inferred posteriors over the decision biases were centered around zero with an SD of about 5 degrees (blue curve in Fig 7B and blue error bars in Fig 7C).

The central motivation for our work is to reward the animal in such a way as to not alter its intrinsic perceptual biases, as demonstrated by our simulations of an RL agent's performance (Fig 3). A useful additional test of our approach is to examine whether an animal's perceptual biases are stable across time. We can do so by comparing the perceptual bias predicted by our stationary, time-independent, linear model (prior mean) with the perceptual bias obtained from the actual responses (posterior mean) (Fig 7C). A diminishing perceptual bias over time would be indicated by a negative trend in the differences between posterior means and prior means across sessions. Conversely, a stable perceptual bias would manifest as fluctuations around a difference of zero. A linear regression analysis on these differences relative to session numbers revealed no significant trends in either leftward ($p = 0.17$) or rightward perceptual biases ($p = 0.68$), consistent with the idea that the perceptual biases remained stable through the course of training with our reward method (Fig 7D).

## Discussion

We propose an adaptive method to reward animals for reporting their subjective (biased) percepts in 2AFC tasks. Our methodology allows neuroscientists to study neural mechanisms of subjective percepts, across long periods of time, without the reward scheme inducing changes in decision criteria that lead to misestimating the true perceptual biases. Our method infers the perceptual bias of the animal separately from other decision-related biases after each trial and rewards the animal based on the estimated perceptual bias. We used a hierarchical Bayesian framework to optimally integrate data from previous sessions, thereby improving the accuracy of bias estimation and reward allocation across training sessions. Using extensive ground-truth simulations, we demonstrated the accuracy and precision of our approach. Simulations of an RL agent demonstrate that conventional reward schemes misestimate perceptual biases due to changes in decision criteria, whereas our method does not. We also applied our method to train a monkey in a motion perception task with unknown subjective percepts, and we show that the monkey's perceptual biases are stable across over 50 training sessions using our reward method, in contrast to results from the same animal in previous studies [8,14]. These findings pave the way for future studies of the neural basis of subjective percepts in animals that require thousands of trials, even when those percepts are not known *a priori* and need to be inferred themselves.

### Rewarding animals when stimulus context biases perception

Several reward strategies have been used in previous work to mitigate the issue of reward allocation in trials for which a perceptual bias is expected (e.g., an illusion). The most straightforward approach involves rewarding animals veridically based on the stimulus [22,23]. Studies employing this method typically train the animal extensively in conditions without perceptual biases. Then, they run limited test trials in the probe (i.e., illusion) conditions for which perceptual biases are expected, under the assumption that rewards will not alter the animal's reports. However, if many probe trials need to be presented, the lack of rewards for specific stimuli incentivizes animals to change their decision criteria to maximize the reward rate (see Figs 2 and 3).

Another strategy used in some prior studies was to give no reward in probe conditions for which a perceptual bias was expected [24–26]. However, given the need for large numbers of trials in neuroscience studies, this strategy could demotivate animals, particularly in tasks (e.g., Fig 6A) for which many conditions are expected to induce perceptual biases. A third strategy is to always reward animals in conditions with perceptual biases [27,28]. This strategy is also problematic for experiments requiring extensive trial counts because animals may learn that they can give any random response in the conditions for which the experimenter wants to measure perceptual biases. A variation on the always reward strategy is to only reward animals after multiple correct trials, the first of which is an illusion trial that is always rewarded [35].

Several other studies employed a strategy intermediate between the previously mentioned strategies: rewarding animals randomly in all [32], 80% [29,30] or 50% [31] of trials with expected perceptual biases, or at a rate proportional to the monkey's performance [33,34]. Again, this strategy could lead to demotivation and random responses when the animal experiences large numbers of trials in conditions that evoke perceptual biases.

Researchers could also reward animals based on a prediction about the expected magnitude and sign of the perceptual biases in their experiment. For example, we could reward animals for reporting motion in world coordinates in the direction discrimination task depicted in Fig 1. However, this strategy might simply "train in" a particular decision-related bias

based on reward feedback, rather than revealing the animal's intrinsic perceptual bias. To avoid introducing artificial decision biases while still motivating the animals, some studies have also tried rewarding animals randomly [8,31] or consistently [15] around their *predicted* perceptual bias. A problem with this approach is that the extent of an animal's perceptual bias typically cannot be predicted *a priori*. Indeed, this method resulted in gradually decreasing perceptual biases over many training sessions in a study involving motion discrimination [8].

In contrast, we propose that the optimal strategy is to infer animals' perceptual biases online and align the reward boundary with the estimated bias. By eliminating the motivation to change decision criteria in order to increase reward, our method ensures that animals continue to report their actual percept, whether they are biased with respect to the physical stimulus or not. Since the algorithm converges rapidly to the true bias – typically within a session – even when the prior is inaccurate (see Fig 5C, dark green), any mismatch between the intrinsic perceptual bias and the reward boundary is short-lived, disappearing before any substantial effect of reward feedback occurs. If no initial bias exists, our method estimates a zero bias and allocates rewards accordingly, without inducing any bias. Our simulations (Fig 3G, 3H) confirm that, under our reward method, perceptual biases remain stable across all conditions, as reward maximization does not necessitate changes in decision criteria. Furthermore, our monkey's perceptual biases remained stable over 50+ training sessions (Fig 7D), consistent with the effectiveness of our strategy. In contrast, data from a previous study [8] involving the same animal found that perceptual biases declined substantially over a similar time period when ambiguous stimulus conditions were rewarded randomly.

## Separating perceptual and decision biases

One of the most challenging aspects of rewarding animals based on their subjective perception is that one needs to distinguish the animal's perceptual biases from other decision-related biases. Since it is impossible to dissociate perceptual from decision-related biases in simple 2AFC tasks [37,49–52], previous studies also interleaved different conditions [36,41–43], presented a variable number of stimuli before the subject responded [37], or used multimodal stimuli [52] to decompose the sources of the empirically measured response bias. Other previous studies intermixed discrimination and estimation tasks trial by trial to measure perceptual biases directly using the estimation task [41,43]. A couple of studies used serial dependencies [38,40] or the tilt surround illusion [39] to separate perceptual from decision-related processes using only estimation tasks. In a recent study, researchers combined all of the aforementioned approaches, and interleaved estimation and discrimination tasks about line lengths while also manipulating the base rate of the stimuli and the visual context with the Muller-Lyer illusion [42]. This allowed the authors to quantify the extent to which the illusion affected perceptual and decision-related processes. Our approach is similar in spirit to methods that interleave conditions with different contexts to elicit different perceptual biases. However, in contrast to the previously mentioned studies, our objective was not to reveal a perceptual process in a specific experiment but to use the inferred perceptual biases to devise a method for rewarding animals to report their subjective percepts in 2AFC tasks in order to study the neural basis of perceptual biases.

## Bayesian methods for estimating psychometric functions

A challenging aspect of rewarding animals based on their subjective perception is that the estimates of biases need to be fairly accurate and stable from the beginning of a training session when only a small number of trials have been completed. Therefore, we developed a hierarchical Bayesian method that can combine data from previous sessions even if some of

the experimental variables, such as eccentricity or heading direction, vary across sessions (see Figs 7 and S1 Fig). This allows us to optimally combine our prior knowledge about the perceptual biases with information coming in from the animal's response after each trial.

Several previous studies have implemented Bayesian inference for estimating psychometric functions in 2AFC tasks [53–57]. However, those previous methods were not designed to infer perceptual biases separately from decision biases but rather to estimate the empirical bias. The primary focus for a subset of these studies was estimating a dynamically changing psychometric function online [55–58], while for other studies the emphasis was on efficient estimation of a more stable psychometric function given all of the responses in the experiment [53,54]. In contrast, we aimed to estimate the perceptual biases of animals separately from other decision-related biases, in an online fashion during the experiment, to reward the animal in real time.

Furthermore, we sought to combine data across sessions for which we needed a method that could apply separate prior distributions over the perceptual and decision biases. In contrast, previous methods (e.g., [53,54]) only allow applying priors to empirical biases. Our approach, which uses separate prior distributions for perceptual and decision biases, proved to be significantly more accurate in estimating perceptual biases than Psignifit [53], a popular off-the-shelf method that only allows for priors over empirical biases. This difference was especially large when the prior over the perceptual biases was stronger than the prior over the decision bias (see Fig 5D, F). Please note that we are not claiming that our algorithm is better at estimating psychometric functions than previous methods. We only claim that using separate priors for perceptual and decision biases results in a more accurate estimation of the biases.

Finally, a previous study proposed an optimal, adaptive training algorithm for animal experiments to speed up the training procedure [58]. Although their method also involves estimating psychometric functions in 2AFC tasks online, in contrast to our method, they aimed to maximize the learning rate of the animals which involved selecting a mixture of easy and difficult stimuli that decrease all biases and history dependencies of the animals [58]. Importantly, their approach assumes that the correct answer is known for each stimulus, and does not address the issue of how one would estimate perceptual biases that are unknown *a priori*.

## Assumptions and limitations

Any method aiming to separately estimate perceptual and decision biases in 2AFC tasks must fundamentally assume that these biases are distinct. In general, the details of how they can be distinguished depend on the specific paradigm employed (e.g. [36,37,52]). In our study, we focus on a paradigm that distinguishes between the two kinds of bias by using contextual modulations. To separately infer decision and perceptual biases, our method makes a critical assumption: the animal must not adjust its decision criteria across contextual conditions more rapidly than our algorithm converges to the true perceptual biases (see S5 Fig C, D). If this condition is violated, the method cannot prevent the development of distinct decision criteria across contexts, which compromises its ability to accurately estimate the animal's perceptual biases (see S5 FigA, S5 FigB). However, decision biases that are shared across contexts will not affect our method's capacity to infer perceptual biases, regardless of how rapidly the decision criterion may change (see S5 FigE, S5 FigF, S5 FigG, S5 FigH).

Our results, along with those from previous studies, indicate that the assumptions underlying our method are satisfied for most animal training studies. First, previous studies suggest that perceptual performance is generally stable within and even across multiple sessions

as changes in perceptual sensitivity typically occur over a slow timescale (5–10 days) [59–69]. Second, empirical evidence suggests that animals only slowly adjust their context-specific decision criteria across multiple sessions to maximize rewards when other reward strategies are used [8,35]. These two factors—the stability of perceptual biases and the relatively slow adjustment of decision criteria—enable our method infer the "true" perceptual bias before animals can adapt their decision criteria to maximize rewards. Furthermore, if biases fluctuate during a session, our method (like most previous methods) will simply return their average values per session.

Decision biases, which are more variable within and across sessions [18–21,70], are unlikely to differ across interleaved contextual conditions. In such cases, our method can reliably estimate perceptual and decision biases separately. Additionally, while our model assumes a stationary decision bias within each session, it remains robust to gradual changes over a few hundred trials (see Fig 5B). Although the most accurate estimate of the average perceptual bias is achieved by weighting all trials in a session equally, the model's robustness to slow nonstationarities suggests that it implicitly underweights earlier trials. This is likely due to approximations in the inference process (see Method section "Inference process") that effectively discount the information from early trials.

Our extended Bayesian model, which integrates data across sessions, is capable of tracking gradual changes in perceptual biases over sessions. This adaptability arises from the fact that it explicitly allows for estimating the session-by-session variability in the biases. As an example, when applied to our monkey data, the prior widths for each session stabilized within a range of 5–10 degrees instead of converging to zero after many sessions. This makes the method with the extended Bayesian model well-suited to accommodate gradual shifts in perceptual biases across multiple sessions, as demonstrated by our simulations (S5 FigC–S5 FigH).

In order to address changes on a shorter timescale, across trials, or tens of trials, our method is adaptable and can be applied alongside other task and decision-making models, including those that account for nonstationarities or sequential effects. A straightforward approach is to apply a sliding window for determining the history of data that is used for estimating the perceptual and decision biases [71]. Alternatively, one can apply a Gaussian process prior over the biases (e.g., [56,58]), or define a function *a priori* capturing the expected temporal evolution of the biases (e.g., [55]). However, these adjustments come with trade-offs. More complicated models with more parameters typically need more data for accurate estimations. Thus a Gaussian process, with its less restrictive prior, or a sliding window approach, which relies on smaller data sets, will likely introduce greater noise into the inference process, potentially diminishing the effectiveness of the method.

In simulations, we found our method to be robust in the presence of sequential choice and reward effects. Specifically, we simulated one type of serial dependency in which animals tend to make the same choice as in the previous trial when it was rewarded, and tend to make the opposite choice when it was not rewarded (i.e., win-stay, lose-switch strategy). Since the contextual conditions are randomly interleaved, the history of a few trials will not have a systematically different effect on the different contextual conditions. Indeed, we found no significant impact on the effectiveness or accuracy of our method (S6 Fig). However, the presence of serial effects increases the uncertainty in the estimates, as serial dependencies violate the model's assumption of trial independence. Nevertheless, this increased uncertainty is moderate and does not significantly impact the accuracy of our method (S6 Fig). Other forms of serial dependencies are expected to produce similar outcomes: preserving the mean of the bias estimates while increasing the uncertainty in those estimates.

In the simulations of Fig 3, the RL agent was given the freedom to independently adjust its decision criterion for each contextual condition, and we showed that our approach prevents changes in decision criteria that would compensate for perceptual biases. Importantly, the extent to which animals can independently adjust their decision criteria across contexts is often unknown. Monkeys' responses exhibited diminishing biases when a random reward strategy was used in trials with contextual modulations [8,35], potentially reflecting adjustments in decision criteria. Another study found that animals adjusted their decision criteria nearly optimally to maximize rewards when relative reward magnitudes were cued in a 2AFC task [18]. However, animals may not always be able to adapt their decision criteria to maximize rewards. For instance, in [72], an animal did not adopt different decision criteria to counteract visual perturbations introduced in trials that were clearly marked by different colors. Thus, the extent to which animals can maintain context-specific decision criteria likely depends on the experimental details. As long as animals adopt different decision criteria sufficiently slowly in multiple sessions, our method will be effective (S5 FigC, D).

Finally, our reward method is designed to track existing perceptual biases; it cannot create a bias that was not already present, nor can it eliminate existing biases. As a result, if empirical biases underestimate perceptual biases due to changes in decision criteria that result from suboptimal reward strategies (e.g., Fig 3D), applying our method cannot recover the true perceptual biases unless the decision criteria somehow relax to a common value across contexts. It might be possible to design additional control conditions, in which the animal's decision criteria under contextual modulations can be estimated. Subtracting these estimates of decision criteria from the empirical bias measurements could help to recover the true perceptual biases.

## Conclusion

Studying the neural basis of perceptual biases poses a substantial challenge in neuroscience. Rewarding animals to report their subjective percepts, which may be different from the presented stimulus, is difficult because the experimenter has no direct access to the animal's subjective percepts. We propose that the best strategy is to infer animals' subjective percepts online from their responses and allocate rewards based on the estimated perceptual bias. We implemented a hierarchical Bayesian framework that can provide a real-time, trial-by-trial estimation of perceptual biases and can also combine data across multiple sessions with variable task conditions. Our simulations of an RL agent confirm that our method outperforms other conventional reward strategies. In addition, data from one monkey trained to perform a motion discrimination task demonstrate stable perceptual biases over many sessions when employing our reward strategy.

## Methods

### Hierarchical Bayesian model

First, we assume that the animal's choice for trial, $t$, denoted as $C_t$, follows a Bernoulli distribution:

$$C_t \sim \text{Bernoulli}(\theta_{t,k}) \tag{1}$$

Second, to model the trial-by-trial decision-making process in the motion direction discrimination task, we model the probability of making the rightward choice, $\theta_{t,k}$, for a stimulus direction, $\omega_t$, in a contextual condition, $k$, using a cumulative Gaussian distribution

function, $\Phi$:

$$\theta_{t,k} = \lambda_{1,k} + (1 - \lambda_{2,k} - \lambda_{1,k})\Phi(\omega_t|B_k, S_k). \tag{2}$$

$B_k$ and $S_k$ correspond to the Gaussian distribution's mean and standard deviation and describe the observer's empirical bias and sensory noise, respectively. Importantly, this probability ($\theta_{t,k}$) is not solely determined by the stimulus on the retina, $\omega_t$, but is also influenced by contextual information in condition $k$, leading to different empirical bias, $B_k$, and sensory noise, $S_k$, parameters in each different contextual condition. We used lapse rates to account for the tendency to make incorrect choices due to guessing, as represented by $\lambda_{1,k}$ and $\lambda_{2,k}$ for leftward and rightward choices, respectively.

The empirical bias, $B_k$, comprises both perceptual, $P$, and decision, $D$, biases in general. In our example with three contextual conditions, we have three empirical biases, $B_L$, $B_N$, and $B_R$, along with three sensory noise variables, $S_L$, $S_N$, and $S_R$, for the leftward, neutral, and rightward self-motion interleaved conditions. The empirical biases in the three contextual conditions are the following combinations of perceptual and decision biases:

$$\begin{aligned} B_L &= P_L + D \\ B_N &= D \\ B_R &= P_R + D \end{aligned} \tag{3}$$

where $P_L$, $P_R$, and $D$ represent the leftward perceptual bias, rightward perceptual bias, and decision bias, respectively. We assume that the perceptual ($P_L$ and $P_R$) and the decision ($D$) biases are stationary within a session (but see Discussion and S5 Fig). We also assume that the decision bias ($D$) does not change across the interleaved contextual conditions.

We assumed Gaussian prior distributions for the perceptual and decision biases:

$$\begin{aligned} P_L &\sim N(P_{0,L}, \tau_{P_L}) \\ P_R &\sim N(P_{0,R}, \tau_{P_R}) \\ D &\sim N(D_0, \tau_D) \end{aligned} \tag{4}$$

where $P_{0,L}$, $P_{0,R}$, and $D_0$ represent the means of the prior distributions, reflecting our best estimates based on prior knowledge obtained before the start of a session (as discussed further below). The standard deviations $\tau_{P_L}$, $\tau_{P_R}$, and $\tau_D$ represent the uncertainty associated with these estimates.

We assumed a gamma prior distribution for the sensory noise:

$$\begin{aligned} S_L &\sim \Gamma(\alpha, \beta) \\ S_R &\sim \Gamma(\alpha, \beta) \\ S_N &\sim \Gamma(\alpha, \beta) \end{aligned} \tag{5}$$

where we used the same parameters, $\alpha$ and $\beta$ in all contextual conditions. Note that if, in other experiments, $\alpha$ and $\beta$ most likely change across the different conditions, our method allows us to use different $\alpha$ and $\beta$ parameters for each experimental condition.

Finally, we assume that all lapse rates $\lambda$ follow a Beta distribution with the same parameters, $\gamma$, and $\epsilon$:

$$\begin{aligned} \lambda_{1,k} &\sim \text{Beta}(\gamma, \epsilon) \\ \lambda_{2,k} &\sim \text{Beta}(\gamma, \epsilon) \end{aligned} \tag{6}$$

### Inference process

We used Gibbs sampling using the STAN Matlab toolbox [73–75] to compute the posteriors over the perceptual and decision biases ($P_L$, $P_R$, and $D$), the sensory noise variables ($S_L$, $S_N$, and $S_R$), and the lapse rates ($\lambda_{1,k}$ and $\lambda_{2,k}$). We generated 5000 samples, of which we discarded the first 2500 (burn-in). In each session, the inference via Gibbs sampling started after the first 33 trials (i.e., after one repetition of all stimulus directions in each self-motion condition). Then, the posteriors over all variables in the model were updated after every subsequent trial. We applied the same Gibbs sampling algorithm to estimate the posteriors both in the simulations and in the monkey training sessions.

The computation time of the inference process depends on the size of the data and the number of samples drawn from the posterior. However, assuming the range of possible data points (500-5000 trials) typical in animal training paradigms, together with drawing 5000 samples from the posterior, our method outputs the best estimates of each bias in 1–2 seconds on an average computer (e.g., 2021 iMac). For monkey training, the algorithm was run on a Dell Precision T3500 desktop computer from 2010 and completed before the monkey made their choice in the next trial (3–3.5 seconds). Finally, computing time can be shortened by reducing the number of samples used to perform the (approximate) computations.

### Reward allocation

Our reward method updates the reward boundary after each trial, $t$, in a session with $T$ trials using the estimated perceptual biases in the three experimental conditions as follows:

1. **Initialization** Defining prior distributions (see Method sections "Hierarchical Bayesian model" and "Extended hierarchical Bayesian model").
2. **Loop through trials:**
   For each trial $t$ from 1 to $T$:
   i. Compute posteriors over biases in each contextual condition, $p(P_L|\omega_{1:t}, C_{1:t})$, $p(P_R|\omega_{1:t}, C_{1:t})$, $p(D|\omega_{1:t}, C_{1:t})$ as follows:
      - If $t < 33$:
         Assume posteriors are equal to their prior distributions.
      - If $t \geq 33$:
         Use data ($\omega_{1:t}$ and $C_{1:t}$) and the Bayesian model (see Method sections "Hierarchical Bayesian model" and "Extended hierarchical Bayesian model") to compute posteriors.
   ii. Compute best estimates of the perceptual biases, $\overline{P}_m$ (where $m = \{L, R\}$ depending on the contextual condition in trial $t$), by taking the mean of the posterior distributions.
   iii. Allocate reward:
      - If ($C_t$ = left and $\omega_t < \overline{P}_m$) or ($C_t$ = right and $\omega_t > \overline{P}_m$):
         Give reward.
      - If ($C_t$ = right and $\omega_t < \overline{P}_m$) or ($C_t$ = left and $\omega_t > \overline{P}_m$):
         Give no reward.

Prior distributions can be based on previous data, prior knowledge, or results from earlier studies when available. In the absence of prior information, priors can initially be set as uniform distributions. During training sessions with monkeys, rewards were often progressively increased to sustain motivation. We believe that the absolute amount of reward does not affect our method. Our method is flexible and can also accommodate alternative reward schemes,

such as a reward escalator, where the reward amount increases based on the number of consecutive correct trials.

## Reinforcement learning with different reward methods

To gain insight into how different reward strategies influence decision criteria and measured biases, we simulated the behavior of a reinforcement learning (RL) agent. Crucially, the RL agent was given the freedom to independently adjust its decision criteria in three interleaved contextual conditions. The RL agent updated its decision criteria on each trial using a temporal difference rule [48]:

$$\psi_{t,k} = \psi_{t-1,k} + \alpha\, C_{t,k}\,(r_{t,k} - q_{t,k})\, \frac{\partial p(C_{t,k}|\omega_{t,k})}{\partial \psi_{t-1,k}} \tag{7}$$

where $\psi_{t,k}$ denotes the decision criterion in trial $t$ for contextual condition $k$. $\alpha$ represents the learning rate, and $r_{t,k}$ denotes the reward (1 if the reward was present and 0 otherwise). The choice in trial $t$ and condition $k$ is $C_{t,k}$, which is –1 for leftward and 1 for rightward choices. Since the agent's "perception" (denoted by $\hat{\omega}_{t,k}$ and defined in eq. 11) remains unchanged during training – only its decision criterion is updated – we approximate the gradient term (where $\phi$ is the standard Gaussian distribution),

$$\frac{\partial p(C_{t,k}|\omega_{t,k})}{\partial \psi_{t-1,k}} = \frac{-1}{\Sigma_k}\phi\left(\frac{\hat{\omega}_{t,k} - \psi_{t-1,k}}{\Sigma_k}\right) \tag{8}$$

by its sign leading to a simplified temporal difference rule:

$$\psi_{t,k} = \psi_{t-1,k} - \alpha\, C_{t,k}\,(r_{t,k} - q_{t,k}) \tag{9}$$

We chose $\alpha = 0.04$ in the simulation to roughly approximate the learning speed in measured behavior. In these simulations, the unit of $\alpha$ is degrees per trial. $q_{t,k}$ represents the predicted probability of receiving a reward in the agent's mind on trial $t$:

$$q_t = \max\left(\hat{\theta}_{t,k}, 1 - \hat{\theta}_{t,k}\right)$$
$$\hat{\theta}_{t,k} = \Phi\left(\hat{\omega}_{t,k}|\psi_{t-1,k}, \Sigma_k\right) \tag{10}$$

where $\Phi$ is the cumulative Gaussian distribution with mean equal to the agent's decision criterion, $\psi_{t-1,k}$, and with standard deviation $\Sigma_k$, evaluated at the agent's stimulus estimate, $\hat{\omega}_{t,k}$. The agent's stimulus estimate was modeled as a noisy observation drawn from a Gaussian distribution centered at the biased stimulus value, $\omega_t - P_{t,k}$, with noise $\Sigma_k = 16$ deg for condition $k$:

$$\hat{\omega}_{t,k} \sim N(w_{t,k} - P_{t,k}, \Sigma_k) \tag{11}$$

Using this model, we ran simulations with the following reward strategies:

1. Rewards were allocated based on the veridical stimulus values (S4 FigA).
2. Rewards were allocated randomly in trials with contextual modulation for which the correct response could not be predicted (i.e., stimulus values fell within the extent of the perceptual bias, as shown by yellow areas in S4 FigB). We refer to such trials as "ambiguous" trials.
3. Rewards were never allocated in ambiguous trials.
4. Rewards were always allocated in ambiguous trials.

5. Rewards were allocated based on the ground truth perceptual biases $P_k$ for each condition $k$ (S4 FigC). Note that this "oracle" strategy is not available to the experimenter since it requires knowledge of the observer's subjective perceptual bias.

6. Rewards were allocated based on the estimated perceptual biases $\overline{P_k}$ obtained using our method.

For each strategy, we simulated 10 independent synthetic training experiments, each consisting of 50 sessions of 990 trials each.

There are two additional assumptions that we made in the RL algorithm detailed above. First, we assumed that the agent follows a maximum a posteriori (MAP) decision strategy, always selecting the choice with the highest probability of being correct, without incorporating decision noise.

Second, the RL algorithm operates under the assumption of a stationary environment. However, when the agent continuously updates its decision criterion while the stimulus distribution remains unchanged, the stimulus statistics shift relative to the evolving decision boundary. This shift results in one choice becoming progressively dominant, creating a self-reinforcing loop that further biases the decision criterion. Over time, this feedback loop causes a bias in the responses, leading to a systematic horizontal shift in the agent's psychometric curve. Notably, this effect persists even when rewards are assigned based on the ground-truth perceptual biases (Fig 3E), contributing to small but measurable changes in decision criteria.

This issue can be mitigated by centering the stimulus distribution on the updated decision criteria (see Fig S3 FigE, S3 FigF). In practice, however, experimenters typically use fixed stimulus sets. To maintain consistency with real-world experimental conditions, we also used fixed stimuli in our simulations. Fortunately, this had only a minor impact on the RL agent's responses, as evidenced by the small effects observed in simulations using the "oracle" strategy (Fig 3E, 3F) and our proposed algorithm (Fig 3G, 3H).

## Simulating gradually changing perceptual biases

To explore gradually changing perceptual biases in S5 Fig, we simulated scenarios in which the bias for condition $k$ evolved across sessions following a piecewise linear function:

$$P_{t,k} = \begin{cases} P_0, & t \leq t_{b1}, \\ P_0 - \frac{(s-1)P_0}{s} \cdot \frac{t-t_{b1}}{t_{b2}-t_{b1}}, & t_{b1} < t < t_{b2}, \\ \frac{P_0}{s}, & t \geq t_{b2}, \end{cases} \quad (12)$$

where $t$, $P_0$, $s$, $t_{b1}$, and $t_{b2}$ denote the trial number, the initial extent of the perceptual bias, the steepness of the change, and the beginning and end of the period of change, respectively. The steepness factor was set to 3 to produce a gradual decrease across multiple sessions and the change starts at the 11th session and ends at the 41st session. To take into account the information across sessions, we used the extended hierarchical Bayesian model (see S1 Fig and Method section "Extended hierarchical Bayesian model"). Now the hyperprior of the perceptual biases are

$$\begin{aligned} P_R &\sim N(P_{0,R}, \sigma_R) \\ P_L &\sim N(P_{0,L}, \sigma_L) \end{aligned} \quad (13)$$

where $P_{0,R}$ and $P_{0,L}$ are the means of the hyperprior, and $\sigma_R$ and $\sigma_L$ are the standard deviations.

## Model validation

We validated our method using ground-truth simulations. We generated synthetic data for a motion direction discrimination experiment with 3 self-motion conditions. The target patch's (dashed inner circles in Fig 6A) directions of motion ranged from -40 to 40 degrees (deg) with steps of 8 deg, resulting in a total of 11 target motion directions. Similar to the real experiment (see Method section "Motion discrimination task with optic flow"), the stimuli in the simulations were generated with block randomization, ensuring that each distinct stimulus condition was shown once before repeating the same set of stimuli in a different randomized order. With three contextual conditions (leftward, neutral, and rightward self-motion), this meant that each block contained a randomized order of 11 (motion directions) x 3 (contextual conditions) trials. In each synthetic session, we generated 30 blocks of trials resulting in 990 trials in total (30 blocks x 11 motion directions x 3 contextual conditions = 990 trials). The monkey's choices were generated using the model described above (Method section "Hierarchical Bayesian model") with different ground truth combinations of perceptual and decision biases. We assumed zero lapse rates in the simulations.

First, in line with the assumptions of our Bayesian model (see Method section "Hierarchical Bayesian model"), we generated data with a stationary decision bias equal to 10 deg (see Fig 5A, 5C, 5E). Second, we generated data with a decision bias that changed slowly within a session following a sinusoidal function, $D = 10 \sin(\pi n/2000)$, where $n$ represents the trial number (see Fig 5B). The ground truth perceptual bias was always 20 deg for the leftward contextual condition (see dashed red lines in Fig 5A, 5B). The perceptual bias in the rightward contextual condition was always –10 deg, which is half the magnitude of the leftward perceptual bias with an opposite sign (see dashed green lines in Fig 5A, 5B). The ground truth value for the sensory noise was consistently set to 15 deg across all conditions and simulations, which approximated the slopes of psychometric functions observed in the monkey behavior during our experiment.

To compute the posteriors, we used the same priors over the sensory noise variables, $S \sim \Gamma(8, 0.5)$, and the lapse rates, $\lambda \sim \text{Beta}(1, 10)$, in all three contextual conditions in each simulation. However, we tested our method with different sets of priors over perceptual and decision biases. We independently generated 100 synthetic sessions for each combination of priors over perceptual and decision biases we tested.

First, we tested how different offsets between the prior mean and the ground truth values of the perceptual bias influence the estimation result (Fig 5C, 5E). We used prior means zero, one, and two standard deviations from the ground truth values for the perceptual biases while fixing the prior over the decision bias to be centered at zero. The standard deviations for perceptual and decision biases were fixed at 5 and 10 deg, respectively. We also included a maximum likelihood estimation version of the model, corresponding to Bayesian estimation with a uniform prior. Since the quantity we are estimating is an angular variable, we chose to use a uniform prior with the range -180 to 180 deg: $P_L \sim U(-180, 180), D \sim U(-180, 180), P_R \sim U(-180, 180)$.

Second, we compared our method to conventional Bayesian methods using the Psignifit library [53], while applying different ratios of prior widths between perceptual and decision biases (Fig 5D, 5F). Thus, we estimated the biases using our method and using Psignifit [53]. The main difference between Psignifit and our method is that one cannot apply priors separately to perceptual and decision biases when using Psignifit—only over the aggregated empirical biases. To make the results of the Psignifit implementation comparable with our method, we used the same priors for the empirical biases for both methods. The prior over the empirical bias in the neutral condition in the Psignifit implementation was chosen to be

equal to the prior over the decision bias in our method, $B_N \sim N(D, \tau_D)$. However, the empirical biases in the leftward and the rightward conditions are sums of perceptual and decision biases. Thus, the priors over the empirical biases in those conditions can be computed by convolving the priors over the decision and perceptual biases. This leads to the following distributions of the empirical biases: $B_L \sim N\left(P_L + D, \sqrt{\tau_{P_L}^2 + \tau_D^2}\right)$, $B_R \sim N\left(P_R + D, \sqrt{\tau_{P_R}^2 + \tau_D^2}\right)$. In the simulations comparing our method to Psignifit, we used the ground truth values for the prior means and the following prior widths for the perceptual and decision biases: (1) $\tau_{P_L} = \tau_{P_R} = 2\sqrt{46}, \tau_D = 4$; (2) $\tau_{P_L} = \tau_{P_R} = 10, \tau_D = 10$; and (3) $\tau_{P_L} = \tau_{P_R} = 4, \tau_D = 2\sqrt{46}$.

Finally, in simulations incorporating sequential choice effects (S6 Fig), we modeled a simple win-stay, lose-shift strategy based on the reward and choice from the previous trial. The subject had a probability, $T_{se}$, of using this strategy, where it would repeat the same choice as the last trial if rewarded, or switch to the alternative choice if not. For the simulation, we chose $T_{se} = 0.25$ as an example, though our method is effective across a range of non-extreme values.

## Motion discrimination task with optic flow

As a test case, we applied our method during the process of training a rhesus monkey (*macaca mulatta*) to perform a motion discrimination task. In this task, the monkey reports the direction of a small patch of dots under 3 contextual conditions involving different surrounding optic flow fields (Fig 6A). While the details of the motivation for this project are beyond the scope of this paper, we provide a brief description of the task here. The task and stimuli are a variant of those recently described in [14], with some key differences outlined below. The experimental apparatus used is identical to that described in [14]. The monkey viewed stimuli that were rear projected onto a display that subtended 90 x 90 deg at a viewing distance of 32 cm. The animal was head restrained, and eye movements were measured and enforced using a scleral eye coil. Visual stimuli were custom generated using OpenGL graphics libraries, and were viewed stereoscopically through red and green filters.

The monkey performed a motion discrimination task in which a patch of random dots moved in a specific direction with 100% coherence (yellow arrows in the dashed inner circles in Fig 6A). After a 2-second stimulus period, the monkey had to indicate their decision by making a saccadic eye movement, within 1 second, to one of two choice targets that appeared after the motion display was extinguished. The direction reference (white dashed line in Fig 6A) was implicitly indicated by the locations of the two saccade targets (not shown in Fig 6A) that were aligned along an axis orthogonal to the direction reference. The monkey received a juice reward for making a saccade to the correct target determined by our online reward method. Across behavioral sessions, the location of the target patch varied extensively, and the direction reference was covaried accordingly, such that the direction reference was always aligned with a ray that extended from the fixation target to the center of the target patch.

We simulated contextual information (mimicking self-motion) using optic flow to investigate whether the monkey's perception of object motion is attracted toward the optic flow vector that would exist at the location of the target patch. Background optic flow filled the entire visual display except for a mask region that was centered on the target patch and roughly twice as large as the target patch (such that background optic flow did not overlap with the target patch's motion). In the Neutral condition, optic flow simulated forward self-motion, and the direction reference for the discrimination task was aligned with the expected optic flow vector at the center of the target stimulus in this condition (white dashed line is aligned

with the blue local motion vector in Fig 6A, center). We did not anticipate any bias in the psychometric curve in this Neutral condition. In the other two conditions, the heading directions (colored circles in Fig 6A, left and right) were shifted along an implicit circle around the screen's center point with a radius equal to the eccentricity of the target patch. The optic flow vectors had the same length in all three conditions, but their angles differed across the contextual conditions (Fig 6A). For the Leftward and Rightward optic flow conditions, the monkey's perception of target direction is expected to be biased to the left (i.e., counter-clockwise) and to the right (clockwise), respectively. Since the two headings simulated by optic flow were always symmetrical around straight forward, we expected perceptual biases in the Leftward and Rightward conditions to be roughly equal in magnitude and opposite in sign.

## Practical implementation of our method in monkey training

The method proposed here was developed in the early stages of a project on motion perception using the task described above. At the beginning of the animal's training, the method described here was not completed. Consequently, the monkey was rewarded veridically during the initial 5 sessions. Then, we introduced a prototype version of the estimation algorithm. This version assumed symmetrical perceptual biases for the leftward and rightward optic flow conditions and independently inferred a perceptual bias for the three conditions. The symmetry assumption was reasonable because the optic flow conditions in our task are symmetrical, and generally produce approximately equal and opposite perceptual biases. After 5 more sessions, we discarded this symmetry assumption and transitioned to inferring all perceptual biases simultaneously, as detailed in section "Hierarchical Bayesian model".

We later transitioned to the extended hierarchical Bayesian model (see section "Extended hierarchical Bayesian model"), and we began updating the priors every few sessions. Since we generally found that the extended Bayesian model predicted priors for the perceptual biases that were nearly symmetrical (e.g., $P_L = -11$, $P_R = +10$), we often split the difference and used symmetrical priors in practice. Also, as a practical fail-safe in the code, we limited changes in the reward boundary between consecutive trials to 1 percent of the stimulus range (equivalent to 1 degree in our task). This safeguard was rarely triggered, occasionally occurring only during the first few updates at the beginning of a session when the prior was misaligned with the eventual estimate of the perceptual bias.

In the two example sessions shown in Fig 6B, 6C, we used priors computed from previous sessions based on the extended hierarchical model (see the next section). In the top subplots in Fig 6B, 6C, we used the following priors: $P_L \sim N(-10, 7)$, $P_R \sim N(10, 7)$, and $D \sim N(0, 10)$. In the bottom subplots in Fig 6B, 6C, we used the following priors: $P_L \sim N(-16, 7)$, $P_R \sim N(16, 7)$, and $D \sim N(0, 10)$. The psychometric curves in Fig 6B show the empirical biases estimated from all trials in the session.

## Extended hierarchical Bayesian model

We extended our model in section "Hierarchical Bayesian model" to incorporate a simple linear model with additional latent variables that capture how the perceptual bias changes with heading direction and eccentricity across sessions (Fig 7B, 7C, 7D, and S1 Fig). We simply assumed that the heading directions and eccentricities have a linear relationship with the perceptual biases:

$$P_R = \boldsymbol{\beta}_R^\top \mathbf{X}$$
$$P_L = \boldsymbol{\beta}_L^\top \mathbf{X}$$

(14)

where $\mathbf{X}$ denotes the task variables that change across sessions, which in our case consist of heading direction and eccentricity, and $\beta$ denotes the weights. Therefore, the prior over the perceptual biases can be written as follows:

$$P_R \sim N(\boldsymbol{\beta}_R^\top \mathbf{X}, \sigma_R)$$
$$P_L \sim N(\boldsymbol{\beta}_L^\top \mathbf{X}, \sigma_L)$$

$$\text{(15)}$$

We used weakly informative hyperpriors (Gaussian distribution centered on zero with SD = 1000) over the variables, $\beta_R$, $\beta_L$, $\sigma_R$, and $\sigma_L$, which were estimated across sessions since we usually have limited prior information for these variables. In S2 Fig, we show the measured relationships between perceptual bias, heading directions and stimulus eccentricity.

In Fig 7C, D, we inferred $\beta_R$, $\beta_L$, $\sigma_R$, and $\sigma_L$, using all 54 sessions of monkey behavioral data. In each session, we use these inferred posteriors over $\beta_R$, $\beta_L$, $\sigma_R$, and $\sigma_L$ (using all sessions) together with the session-specific values of $\mathbf{X}$ to construct the session-specific priors over the perceptual and decision biases. Then, we estimate the posteriors (y-axis, Fig 7C) over the perceptual and decision biases using these session-specific priors (x-axis, Fig 7C).

## Ethics statement

Behavioral data were collected, using our algorithm to determine how the animal was rewarded on each trial, from one rhesus monkey (Macaca mulatta). The animal was surgically prepared for experiments by implanting a ring for head restraint as well as a scleral coil for measuring eye movements. All animal surgeries and experimental procedures were approved by the University Committee on Animal Resources at the University of Rochester (approval #100682). Surgical procedures were performed under isoflurane anesthesia using a sterile technique. Peri- and post-operative analgesics were provided, in consultation with veterinary staff from the Division of Comparative Medicine, to ameliorate pain and suffering.

## Acknowledgments

The authors would like to thank Brandon Rabah, Bridget Williams, and Hayley Brigham for their assistance with monkey training; Zhe-Xin Xu for contributing to stimulus programming in the monkey task; and Tanzy Love for providing support with Stan coding.

## Supporting information

**S1 Fig. Extended generative model with hyperpriors.** The two inner plates represent the same generative model as in Fig 4C. This model is extended by incorporating latent variables shared across sessions. The vector $\vec{\beta}$ encodes the weights that determine how perceptual biases vary with heading direction and stimulus eccentricity across sessions. $\sigma_P$ and $\sigma_D$ represent the standard deviations of the perceptual and decision bias random variables, respectively. See Method section "Extended hierarchical Bayesian model" for further details.
(TIFF)

**S2 Fig. The relationship between perceptual biases, heading direction, and object eccentricity.** (A) Relationship between perceptual bias and the heading direction simulated by optic flow in the monkey task. With a larger heading direction, the monkey showed a larger perceptual bias in leftward and rightward conditions. Symbol colors, from light to dark, represent eccentricity, from large to small. (B) Relationship between perceptual bias and object eccentricity. With a larger eccentricity, the monkey showed a larger perceptual bias in leftward and rightward conditions. Symbol colors from light to dark represent

heading directions from large to small. Source data for S2 FigA-B are available in "analysis_data/Figure7BCD&FigureS2.mat" at https://doi.org/10.5281/zenodo.15341390.
(TIFF)

**S3 Fig. Additional RL agent simulations.** (**A, B**) Same as (Fig 3A, 3B), but with an "always reward animals in ambiguous trials" strategy. (**C, D**) Same as Fig 3A, 3B, but simulating a "never reward animals in ambiguous trials" strategy. Note that results for both the "always" and "never" reward strategies are quite similar to those of the random reward strategy shown in Fig 3C, 3D. (**E, F**) Results from an RL agent simulation in which reward is based on the ground truth perceptual biases, as in Fig 3E, 3F. The only difference is that, in this simulation, the range of object directions was symmetrical around the true perceptual bias for each of the contextual conditions specified by optic flow. In this case, decision criteria remain the same across contexts, indicating that the small separation observed in Fig 3E results from stimulus range effects. Source data for S3 FigA– S3 FigF are available in "simulation_data/Figure3&FigureS3.mat" at https://doi.org/10.5281/zenodo.15341390.
(TIFF)

**S4 Fig. Diagram of choice outcomes for different reward paradigms.** The blue curve represents an example psychometric curve, with perceptual bias $P = -20$ deg and slope $S = 16$. Shading indicates the proportions of trials that are scored as correct (green) or incorrect (red), as well as trials that are rewarded differently in different methods (yellow). (**A**) Rewards are based on veridical stimulus value, such that the reward boundary is at zero object direction despite the perceptual bias. (**B**) Yellow shading indicates a range of stimulus values for which it is assumed that the "correct" answer cannot be known (i.e., there is an illusion). Within this range, rewards are delivered randomly, always, or never. (**C**) Rewards are based on the animal's subjective percept, such that the reward boundary is aligned with the true perceptual bias of $P = -20$ deg. Note that the animal will receive more total reward (less red area) as compared to the veridical reward strategy.
(TIFF)

**S5 Fig. RL agent simulations for slowly-changing perceptual biases.** Each panel shows 50 simulated sessions, with each session comprising 990 trials. (**A, B**) The RL agent adjusts independent decision criteria for each optic flow condition, with a moderately fast learning rate of 0.04 deg/trial. (A) Solid curves depict the learned decision criteria across the three contexts: leftward (red), rightward (green), and neutral (blue) self-motion. The dashed black line (barely visible behind the solid blue line) represents zero decision criterion. (B) Solid curves show the estimated empirical biases in the three self-motion conditions. Red and green dashed lines show how the ground truth perceptual biases change over time, starting at +20 and -10 deg for leftward and rightward self-motion, respectively, and decreasing linearly to 20/3 and –10/3 deg before stabilizing. In this case, the RL agent could adjust its decision criteria faster than our method could track the changing perceptual biases, causing a mismatch between the measured (solid) and true (dashed) biases. (**C, D**) Same as (**A, B**), but with the RL agent having a much slower learning rate of 0.0004 deg/trial. In this case, our method accurately tracks the changing perceptual biases. (**E, F**) Same as (**A, B**), but with an RL agent restricted to having a single decision criterion that is shared across contextual conditions. Since our method estimates perceptual biases independently of any decision biases that are shared across conditions, it performs effectively in this scenario. (**G, H**) Same as (**E, F**), but with a much faster learning rate of 4 deg/trial. If the decision criterion is shared across contexts, our method works well even with very fast learning

rates. Source data for S5 FigA–S5 FigH are available in "simulation_data/FigureS5.mat" at https://doi.org/10.5281/zenodo.15341390.
(TIFF)

**S6 Fig. Assessing our method's robustness to the presence of sequential choice effects.** (**A**) A ground-truth simulation very similar to Fig 6A, 6B, but with built-in sequential choice effects. On each simulated trial, the agent is given a 25 % probability of using a win-stay-lose-shift strategy, repeating the previous choice when rewarded and switching to the alternative choice when not rewarded, regardless of the stimulus values. Note that our method still tracks the ground-truth perceptual biases in the presence of the choice history effect. (**B**) Solid curves: Average root mean square error (RMSE, y-axis), across 20 simulations, in estimating perceptual bias in the rightward self-motion condition, plotted as a function of trial number. Results are shown for three different prior mean values: 0, 1, and 2 standard deviations (SDs) away from the ground truth perceptual bias (from light to dark green, respectively). Dashed curves: 20 simulations replotted from Fig 5C (matching the repetition number used for the solid lines), which didn't include any sequential choice effects. When a sequential effect is present, the RMSE shows a slight increase but remains relatively small, indicating that the model's performance is still adequate. (**C**) Analogous result to panel B, but showing the average standard deviation (SD) of perceptual bias estimates as a function of trial number. Again, performance is similar, but SDs are somewhat larger in the presence of choice history effects. Source data for S6 FigA–S6 FigC are available in "simulation_data/FigureS6A.mat" and "simulation_data/FigureS6BC.mat" at https://doi.org/10.5281/zenodo.15341390.
(TIFF)

## Author contributions

**Conceptualization:** Yelin Dong, Gabor Lengyel, Sabyasachi Shivkumar, Ralf M Haefner, Gregory C DeAngelis.

**Data curation:** Yelin Dong, Gabor Lengyel, Grace F DiRisio.

**Formal analysis:** Yelin Dong, Gabor Lengyel, Sabyasachi Shivkumar.

**Funding acquisition:** Ralf M Haefner, Gregory C DeAngelis.

**Investigation:** Yelin Dong, Gabor Lengyel, Sabyasachi Shivkumar, Grace F DiRisio, Ralf M Haefner, Gregory C DeAngelis.

**Methodology:** Yelin Dong, Gabor Lengyel, Sabyasachi Shivkumar, Ralf M Haefner, Gregory C DeAngelis.

**Project administration:** Gabor Lengyel, Ralf M Haefner, Gregory C DeAngelis.

**Resources:** Gregory C DeAngelis.

**Software:** Yelin Dong, Akiyuki Anzai.

**Supervision:** Gabor Lengyel, Ralf M Haefner, Gregory C DeAngelis.

**Validation:** Yelin Dong, Gabor Lengyel.

**Visualization:** Yelin Dong, Gabor Lengyel.

**Writing – original draft:** Yelin Dong, Gabor Lengyel, Ralf M Haefner, Gregory C DeAngelis.

**Writing – review & editing:** Yelin Dong, Gabor Lengyel, Sabyasachi Shivkumar, Ralf M Haefner, Gregory C DeAngelis.

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
