## [Editor Report · Decision Letter 0]

22 Jul 2024

Dear Dr Lengyel, 

Thank you for submitting your manuscript entitled "How to reward animals based on their subjective percepts: A Bayesian approach to online estimation of perceptual biases." for consideration as a Methods and Resources by PLOS Biology.

Your manuscript has now been evaluated by the PLOS Biology editorial staff as well as by an academic editor with relevant expertise and I am writing to let you know that we would like to send your submission out for external peer review.

Once your full submission is complete, your paper will undergo a series of checks in preparation for peer review. After your manuscript has passed the checks it will be sent out for review. To provide the metadata for your submission, please Login to Editorial Manager (https://www.editorialmanager.com/pbiology) within two working days, i.e. by Jul 24 2024 11:59PM.

Kind regards,

Christian

Christian Schnell, PhD

Senior Editor

PLOS Biology

cschnell@plos.org

---

## [Decision Letter · Decision Letter 1]

2 Oct 2024

Dear Dr Lengyel,

Thank you for your patience while your manuscript "How to reward animals based on their subjective percepts: A Bayesian approach to online estimation of perceptual biases." was peer-reviewed at PLOS Biology. I am sorry for the long delay in sending our decision, caused by delays in the review process and me attending a conference last week. Your manuscript has now been evaluated by the PLOS Biology editors, an Academic Editor with relevant expertise, and by several independent reviewers. 

In light of the reviews, which you will find at the end of this email, we would like to invite you to revise the work to thoroughly address the reviewers' reports.

As you will see below, the reviewers think that the study is very well executed and provides important insights. However, they also raise a number of concerns, that are mostly addressable through additional analyses and textual clarifications. 

Given the extent of revision needed, we cannot make a decision about publication until we have seen the revised manuscript and your response to the reviewers' comments. Your revised manuscript is likely to be sent for further evaluation by all or a subset of the reviewers.

**IMPORTANT - SUBMITTING YOUR REVISION**

*Re-submission Checklist*

*Published Peer Review*

*PLOS Data Policy*

*Blot and Gel Data Policy*

Sincerely,

Christian

Christian Schnell, PhD

Senior Editor

PLOS Biology

cschnell@plos.org

REVIEWS:

Reviewer #1: In their manuscript "How to reward animals based on their subjective percepts: A Baysian approach to online estimation of perceptual biases", Dong, Lengyel and colleagues have developed an elegant method to update the reward for each choice in a perceptual decision task based on the estimated perceptual bias of the animal. The manuscript is relevant to the important question about the biological nature of perceptual biases and how we can study these in animals. The proposed reward method is grounded in Bayesian methods, highly principled and constitutes an important improvement on current methodology. However, it relies on number of prior assumptions about the nature and dynamics of perceptual and decision biases and how they can be measured. The method will be very useful to the community, but I think this manuscript is a missed opportunity to explore more fundamentally the neurobiological nature of the different biases in perceptual processing. 

1) Perceptual biases are important because they shift the most sensitive parts of the psychometric functions in different contexts, allowing the animal to maintain discrimination performance. If you so will, they allow to probe with heightened sensitivity along a spectrum. The goal of the introduced reward method is to counteract a diminishing perceptual bias when the animal is not rewarded for reporting it over many trials and sessions. A phenomenon the authors have reported before. With their Bayesian approach to rewarding the animal's choices, the authors try to stabilise the accurate report of a perceptual bias. What is not addressed is whether their method stabilises the report of a persisting perceptual bias or whether their reward schedule is necessary to maintain the perceptual bias itself. This question is important for our understanding of the neurobiological basis of perceptual biases and the effects of reward on perception. It also has implications for the reward method introduced here and how it can and should be applied.

If the Bayesian reward method simply stabilises the report for a perceptual bias that persists, then the animal can shift their behavioural response criterion independently for different trials in different directions in the leftward and rightward bias inducing contexts. This contradicts at least some reports of what animals can do when different reward criteria are explicitly signalled by a change in the stimulus (e.g. Cicmil, Cumming, Parker, Krug 2015, elife, visual perturbation experiment). However, if this is indeed the case, how reliable is then report from the prior or current sessions to infer perceptual and decision biases and how robust is the method in supporting the "real" percept when the experimenter gets it qualitatively not just quantitatively wrong (e.g. no bias if there is one)?

In contrast, if the reward schedule is necessary to maintain the perceptual bias itself, then the new reward method potentially actively alters visual processing of the stimulus. In this context, the literature showing effects of expected reward on neuronal signalling in visual perceptual decision tasks is relevant (e.g. Baruni, Lau, Salzman 2015 Nature Neuroscience; Cicimil, Cumming, Parker, Krug 2015 elife) and should be discussed. 

An important experimental test would be whether - using this new Bayesian reward method - the report of a perceptual bias in an animal in which the report of a perceptual bias has disappeared over time can be re-established. If the method does what it is proposed to do according to the authors, then this reward schedule would recover a persistent but unreported perceptual bias. But if the perceptual bias itself has disappeared, then the dynamic updating of the reward schedule should not induce a new perceptual bias.

2) There a number of assumptions here in order for the reward schedule to work.

All biases are stationary within a session. There is plenty of evidence that this is not the case; the authors cite some themselves. Can the authors estimate explicitly up to what point their model is robust, in other words what temporal variability in the bias they can tolerate?

One bias that is ignored are sequential effects in decision-making, e.g. Lueckman, Macke, Nienborg 2018 J Neuroscience. Reward history has been implicated as a factor in this. The authors should discuss potential interactions with the effects of their reward schedule.

There need to be two contexts that induce perceptual biases in equal and opposite directions. How can the method be adapted of the perceptual bias is just in one direction or might be unequal?

3) There are a number of points where the methodological choices for determining reward trial-by-trial could be more explicitly discussed for the non-expert reader.

The manuscript (Figure 2) states that effectively the reward boundary is shifted (trial-by-trial) and then the animal's choice rewarded depends on which side of the boundary the response falls on for a given stimulus and cortex. Why is there not an element of probabilistic reward near the criterion, because near-threshold choices would not yield choices all in one direction? Would it potentially improve the outcome if the reward probability was explicitly altered?

It would be helpful to see in the methods a full equation that calculates the reward boundary and how it is updated trial-by-trial, rather than just the individual parts of the equations spread out. 

There seems to be no element of discounting choices on trials earlier in the session through differential weigthing as the session progresses. Is this an active choice and why? 

Was the amount of reward per trial flat throughout the experiment and the simulations? How would the outcome be affected by a reward escalator typically used when working with monkeys, i.e. where the amount of reward is determined by the number of preceding correct trials?

Minor points:

- figure 5A: the panel and elements are to small to see adequately on a printed out version. Online magnfication reveals poor resolution of, for example, the white elements in the receptive field.

- figure legend for Figure 5C refers to a "fat" prior. Should be flat.

- the "flat" prior is not defined in the manuscript

- detailed methods on the visual stimulus and task (luminance, size of stimulus, eye tracking etc.) would be useful (or a reference to another paper in the methods specifying these parameters).

Reviewer #2: Dong and colleagues proposed a new method of adaptively rewarding monkeys for experiments involving perceptual biases, based on online estimates of the perceptual bias. They used a Bayesian method to obtain an estimate of perceptual bias for each trial, incorporating and updating a prior that was initially derived from previous sessions, and set the reward boundary that combined the actual stimulus info and the estimated perceptual bias. They used simulations to demonstrate the robustness of their online estimation method and showed that, in one monkey, training with this new reward method did not induce a gradual reduction in perceptual bias. 

This study addresses an important issue in studies of animal cognition, where it is often difficult to interpret changes in a behavioral report in relation to changes in perception alone. The manuscript is well-written and the analyses were performed rigorously. The results are convincing for estimating perceptual bias. However, I am not fully convinced that the online-estimate-based reward method is necessary in practice. 

Major comments

- This study was motivated by a concern that existing rewarding methods might cause apparent decreases in perceptual bias because an animal could in theory learn higher-order strategies to compensate for the biasing conditions. As an example, the authors cited one set of results from their previous experiment (Fig 11 of Peltier et al 2020), stating that the perceptual bias decreased over time. However, I am not sure that this is an accurate description of the figure. For example, during training, monkey M increased its flow-parsing gain and maintained that gain through >30 sessions (estimated by eye; experiments 1 and 2). The gain was only decreased, quite abruptly, when a new experimental condition was introduced and then was maintained for the rest of the study. Monkey P decreased the gain during training, but then maintained it over a year without further reduction after the initial decrease. These results do not unequivocally support the authors' concern about a gradual decrease in perceptual bias with the existing reward methods. 

- The authors' ground-truth simulations showed convincingly that they can estimate perceptual bias online and their estimation method outperforms Psignifit. However, given that the goal of the paper is to devise a reward method to counter compensatory adjustments in decisions, it is surprising that the authors did not try to recapitulate the learning process in their ground-truth simulations to demonstrate how their method compares with alternative methods (e.g.., random rewards, a fixed boundary at the objective 0, a fixed boundary based on the informed prior, online estimates based on some wrong values of prior mean/sd, and online estimates based on Psignifit, etc.). It would also be helpful to understand better how much errors in perceptual bias estimates propagate over trials/sessions through suboptimal reward boundaries. 

- For simulations, it was assumed that the decision bias D was constant from trial to trial with an interleaved-context design. However, this raises the question of why/whether monkeys would still be able to compensate for perceptual biases with such a task design. How much of the stable behavior is due to the interleaved-trial design versus the reward method?

- The results in Figure 5 need clarifications. First, for the lower example, there seems to be a substantial discrepancy between the decision bias (near zero, estimated by looking at the blue psychometric curve) and those in C. How much does such a discrepancy matter? Second, it is not clear why flat prior has higher variability than informative prior only for the rightward condition. Third, how much of the dynamics in perceptual and decision bias were due to estimation versus monkey changing its behavior based on the reward method?

- Figure 6 is supposed to support the idea of pooling multiple sessions to get better prior estimates. However, while panel A shows that a larger prior width is beneficial (the mean bias estimate is closer to ground truth at larger prior width), panel B shows the SD decreases with increasing number of sessions included. This apparent contradiction is likely due to better estimates of *mean* prior with more sessions, which is unfortunately not shown.

- I appreciate the practical difficulty of testing different reward methods on monkeys. However, it remains that the data from 50 sessions in one monkey are not strong evidence that the critical determinant is the new reward method (especially given that monkeys may also maintain stable biases with existing methods, e.g., monkey M in Peltier et al. 2020). 

Minor:

- At the beginning of a session, the authors gathered data from the first 33 trials. How were these trials rewarded?

- Line 285, the white dashed line in Figure 5A was not visible. 

- Pg 12, Figure 5 legend. line 5 from bottom. "fat" -> "flat"?

Reviewer #3: This paper describes a novel psychophysical modeling framework for separating out perceptual and decision biases in psychophysical experiments, along with a closed-loop method for rewarding animals based on their percepts (while continuing to penalize them for decision-related bias). This overcomes an important limitation of previous work, which (if it relies on rewarding animals purely on the basis of "ground truth" stimuli) will effectively train an animal to over-ride those perceptual biases that scientists would like to analyze. The paper provides an extremely elegant solution to a compelling problem. I enjoyed it immensely and feel it will make an outstanding contribution to the literature. I have only two general comments and a number of more detailed comments for improving readability and clarity.

General comments:

---------------

1. Given that the method needs to run quickly enough to provide an updated estimate of the perceptual bias during the inter-trial interval, I think it's important to show some kind of speed comparison. How long does it take to run (and how does it scale with dataset size?) (Apologies if this was mentioned somewhere in the paper, but I did not see it!)

2. A second comment is that I feel the authors have missed an opportunity to show the superiority of their method in a setting where the animal updates its decision strategy based on the rewards it receives. The paper's motivation (eg made explicit in Fig 2B) leans heavily on the idea that rewarding an animal based on the "veridical" or "world" stimulus will lead it to learn to override its perceptual biases and simply make reports that maximize its probability of reward. (In other words, it will learn to give unbiased responses). However, the simulations did not consider any scenario where this occurs; in the simulated datasets, the animal had fixed values of the perceptual biases P_R and P_L and either fixed or sinusoidally modulated values of the decision bias D.

It seems to me that it would be a stronger validation to consider a setting where the animal updates its decision parameter according to some kind of reinforcement learning rule to maximize the reward. (For example, D could be updated according to a policy gradient learning rule... or perhaps some kind of Q learning?). One could then explicitly compare the efficacy of the different reward strategies proposed: rewards based on true stimulus, random rewards or no rewards for stimuli that elicit biased percepts, or rewards based on the authors' proposed method. This would also allow them to show that their method leaves the animal's original perceptual bias more intact than alternative methods!

(Note I recommend adding such a simulation, but I would not consider it essential). 

Detailed comments:

----------------

Sec 3.1 : I'm afraid it isn't clear enough what the task is. (What is the "vertical task reference"? Is this a separate task? It isn't clear what this is or how it relates to the standard task.). All the text says about the task is that it is "object motion perception"; please provide a bit more detail on what the stimulus is and what the animal is expected to do. It also isn't clear to me why retinal velocity and world velocity are different. (If the observer isn't actually moving -- i.e., there are just some distractor dots in the background -- then isn't the retinal velocity the same as the world velocity). The text here feels as though it is written for a reader who is already intimately familiar with the experimental details from some earlier study.

-----

line 168: "The two perceptual biases can then be computed by subtracting the empirical bias measured in the neutral condition from the empirical biases measured in the other two conditions (Fig. 3B)."

Does this mean you DON'T assume that the two perceptual biases have equal magnitude but opposite sign (i.e., the critical assumption made by the first method?). It would be nice to spell this out, if so. (Also, it would be nice if you follow up later to state whether this assumption holds or not in real data!)

Edit: after reading the following section (which describes the details of the method), I wouldn't describe your method in quite this way. If I understand correctly, it's not that you're using just the neutral condition trials to estimate the get an estimate of the decision-bias and then subtracting it off; you're using data from all trials to infer all the model parameters, right? (This is actually a good thing from a Bayesian perspective-- better than separately trying to estimate decision bias from only 1/3 of the trials). So this is maybe the essence of what's happening, but it might be worth clarifying that this isn't literally the approach you're taking.

Edit #2: after reading 3.3 I see that indeed you don't assume the two biases are symmetric. So it might be worth pointing that out above when you introduce the method!

-----

185: First, as commonly done (Prins, 2023;

186 Schutt et al., 2016), we assume that the percentage of making one of the two choices (e.g., a

187 rightward response) follows a binomial distribution with parameter theta denoting the probability

188 of choosing the first response alternative, and parameter n representing the number of trials.

I might suggest describing this as Bernoulli on each trial (instead of binomial over n trials). Here n isn't really a 'parameter' of the model in the sense of parameters you're fitting.

By the same token, I wouldn't refer to theta as a parameter either. (Yes, it's the parameter of the Bernoulli distribution on each trial, but it's not a parameter you're fitting to the data, but rather a quantity derived from the stimulus and model parameters on each trial... you might call it a latent variable). So I would advise being a bit more careful with terminology and definitions. (And as I note below, adding a few equations would also help!)

-----

189: we used a

190 cumulative Gaussian distribution as the functional form of the psychometric curve, reflecting

191 the relationship between ✓ and the stimulus value, !, (e.g., object motion direction).

Personally I would recommend putting the actual equation in here (which I think is easier / clearer than trying to say this in words), e.g., 

P(right choice | omega) = theta = Phi ( (omega - B) / S),

where B = P + D, and Phi is the cumulative normal distribution function. (Or whatever you have in mind).

There's no need to reserve all equations for the Methods section, in my view!

(Note that the graphical model in Fig 3C is not as helpful as having a simple equation!)

-----

208 At the beginning of each session, we gather data from the first 33 trials, each

209 containing one data point from all unique stimuli.

How do you reward the animal during these initial 33 trials?

-----

209 Using Bayesian updating, we combine the

210 behavioral choices from these trials with our prior beliefs to compute an initial estimate of the

211 posterior distribution over perceptual and decision biases. Then, we update our estimates of the

212 posterior distributions over each bias based on the animal's response in each subsequent trial.

213 This way, we can estimate the perceptual bias of the animal trial-by-trial and flexibly update

214 the reward boundary based on the estimated perceptual bias (see Methods 6.1 for more details

215 and a formal description of the model).

I appreciate that the details are in the Methods, but once again I feel it would be useful to have a little bit more detail in the main text. How do you do inference? (Are you running optimization to get a MAP estimate, or using MCMC to get samples?).

I would also expect some discussion of speed -- clearly the method needs to run fast enough to be able to compute an updated estimate of the parameters during the intertrial interval. Is your implementation fast enough for this to be the case? (How does compute time scale with the amount of past data?)

-----

245 In contrast, employing a Bayesian method from an o↵-

246 the-shelf library confines one to assigning priors solely to the empirical biases. Therefore, we

247 assessed under what circumstances and to what extent our method outperforms the conventional

248 Bayesian approach, as implemented using the Psignifit library (Schutt et al., 2016).

I'm sorry, I didn't follow this. If Psignifit only allows you to infer empirical biases, how do you extract estimates of the perceptual bias?)

Likewise, I was puzzled by Fig 4 panels D and F. At first it looked to me like "Our Method: Psignifit: " was the title for these two plots. (It took me a while to realize these were referring to the legend for the line traces below them). But it's also unclear from the fig or caption what it means to be comparing "your method" to Psignifit. (Do you want to give your method a name?)

-----

Methods:

564: We used lapse rates...

This is great. I was actually thinking while reading the paper that I would like to ask you to perform some validation that the method was robust to lapses. I didn't realize until I got here that your model already incorporates lapses. So let me put in another appeal to put equation for theta_m,k into the main text. (Most readers won't bother reading the Methods section, so I think it's worth spelling out the main details of the model in the Results section!). There's no need to give all the prior distributions, but I think the basic descriptive model as well as the formulas for B_L, B_N, and B_R.

Note however that I am a bit confused because the first equation in Sec 6.1 describes the choice probabilities in terms of the cumulative normal distribution, without lapses. So is this a separate model from the one described above? Did you run some versions with lapses and others without? The two equations seem to contradict each other in the current Methods section -- please clarify!

Also, please number all equations!

---

## [Decision Letter · Decision Letter 2]

7 Apr 2025

Dear Gabor,

It was nice to meet you and chat to you in person last week at Cosyne!

Thank you for your patience while we considered your revised manuscript "How to reward animals based on their subjective percepts: A Bayesian approach to online estimation of perceptual biases." for publication as a Methods and Resources at PLOS Biology. This revised version of your manuscript has been evaluated by the PLOS Biology editors, the Academic Editor and one of the original reviewers.

Based on the reviews and on our Academic Editor's assessment of your revision, we are likely to accept this manuscript for publication, provided you satisfactorily address the following data and other policy-related requests:

* We would like to suggest a different title to improve its accessibility for our broad audience: 

"A Bayesian approach to online estimation of perceptual biases helps to tailor rewards to the individual animal"

* Please add the links to the funding agencies in the Financial Disclosure statement in the manuscript details.

* DATA POLICY:

Regardless of the method selected, please ensure that you provide the individual numerical values that underlie the summary data displayed in the following figure panels as they are essential for readers to assess your analysis and to reproduce it: 7A 

* CODE POLICY

* Please note that per journal policy, we do not allow the mention of "data not shown", "personal communication", "manuscript in preparation" or other references to data that is not publicly available or contained within this manuscript. Please either remove mention of these data or provide figures presenting the results and the data underlying the figure(s).

We expect to receive your revised manuscript within two weeks. 

*Published Peer Review History*

*Press*

Sincerely,

Christian

Christian Schnell, PhD

Senior Editor

cschnell@plos.org

PLOS Biology

Reviewer remarks:

Reviewer #1: I would like to thank the authors for answering my questions with such clarity. I am content with the clarifications and additional analyses in the manuscript.

---

## [Editor Report · Decision Letter 3]

29 Apr 2025

Dear Gabor,

Thank you for the submission of your revised Methods and Resources "Rewarding animals based on their subjective percepts is enabled by online Bayesian estimation of perceptual biases." for publication in PLOS Biology. On behalf of my colleagues and the Academic Editor, Adam Kohn, I am pleased to say that we can in principle accept your manuscript for publication, provided you address any remaining formatting and reporting issues. These will be detailed in an email you should receive within 2-3 business days from our colleagues in the journal operations team; no action is required from you until then. Please note that we will not be able to formally accept your manuscript and schedule it for publication until you have completed any requested changes.

When you attend to those requests to come, please also make sure to add a statement to each figure legend where the corresponding source data can be found.

PRESS

We frequently collaborate with press offices. If your institution or institutions have a press office, please notify them about your upcoming paper at this point, to enable them to help maximize its impact. If the press office is planning to promote your findings, we would be grateful if they could coordinate with biologypress@plos.org. If you have previously opted in to the early version process, we ask that you notify us immediately of any press plans so that we may opt out on your behalf.

Sincerely, 

Christian

Christian Schnell, PhD

Senior Editor

PLOS Biology

cschnell@plos.org